



# Missing connectivity during summer drought controls DOC mobilization and export in a small, forested catchment

Katharina Blaurock[1], Burkhard Beudert[2], Benjamin S. Gilfedder[1], Jan H. Fleckenstein[1 3], Stefan Peiffer[1], Luisa Hopp[1]

[1]Department of Hydrology, Bayreuth Center of Ecology and Environmental Research (BayCEER), University of Bayreuth, Bayreuth, 95447, Germany
[2]Department of Nature Conservation and Research, Bavarian Forest National Park, Grafenau, 94481, Germany
[3]Department of Hydrogeology, Helmholtz Centre for Environmental Research, 04318, Germany

*Correspondence to*: Katharina Blaurock (katharina.blaurock@uni-bayreuth.de)

**Abstract.** Understanding the controls on event-driven DOC export is crucial, as DOC is an important link between the terrestrial and the aquatic carbon cycles. We hypothesize that topography is a key driver of DOC export because it influences hydrologic connectivity, which can inhibit or facilitate DOC mobilization. To test this we studied the mechanisms controlling DOC mobilization and export in the Große Ohe catchment, a forested headwater in a mid-elevation mountainous region in Southeastern Germany. Discharge and stream DOC concentrations were continuously measured using in-situ UV-Vis

spectrometry from June 2018 until October 2020 at two topographically contrasting sub-catchments of the same stream: at a steep hillslope (888 m.a.s.l.) and in a flat and wide riparian zone (771 m.a.s.l). We focus on four events with contrasting antecedent hydrological conditions and event size. During events, in-stream DOC concentrations increased up to 19 mg L$^{-1}$ in comparison to 2 - 3 mg L$^{-1}$ during baseflow. The concentration-discharge relationships exhibited pronounced but almost exclusively anti-clockwise hysteresis loops, which were generally wider in the lower catchment than in the upper catchment

due to a delayed DOC mobilization in the flat riparian zone. The riparian zone released considerable amounts of DOC, which led to a total DOC load up to 522 kg per event. The total DOC load increased with the total catchment wetness. We found a disproportionally high contribution to the total DOC export of the upper catchment during events following a long dry period. We attribute this to the lack of hydrological connectivity in the lower catchment during drought, which inhibited DOC mobilization, especially at the beginning of the events. Our data show that not only event size but also antecedent hydrological

conditions strongly influence the hydrological connectivity during events, leading to a varying contribution to DOC export of different catchment parts depending on topography. As the frequency of prolonged drought periods is predicted to increase, the relative contribution of different catchment parts to DOC export may change in the future, when hydrological connectivity will occur less often.



## 1 Introduction

The hydrologic and carbon cycles are tightly coupled, as terrestrial systems store and release carbon into aquatic systems, which act as a carrier for carbon through landscapes. Inland waters may influence the global carbon cycle more than previously thought (Battin et al., 2009). Dissolved organic carbon (DOC) export from streams is an important link between the terrestrial

and the aquatic carbon cycle and a crucial component of the net ecosystem carbon balance (Kindler et al., 2011). The global DOC export reaching inland waters annually from the terrestrial environment is estimated at 5.1 Pg C. A large part (3.9 Pg C) outgasses to the atmosphere in form of the greenhouse gases $CO_2$ or $CH_4$ (Drake et al., 2018). As DOC is converted to these greenhouse gases, it plays an important role in the context of climate change.

DOC concentrations in freshwater systems usually vary from 1 to 10 mg $L^{-1}$ in streams and lakes but can reach up to 60 mg $L^{-1}$

in swamps and bogs (Thurman, 1985). Since the beginning of the 1980s, an increase in DOC concentrations has been observed in a large number of streams, rivers and lakes of the Northern hemisphere (Evans et al., 2005; Roulet and Moore, 2006; Monteith et al., 2007; Freeman et al., 2001). Rising DOC concentrations indicate an increased leaching from soils and peatlands and therefore influence the terrestrial carbon pools, which are of global importance for carbon storage (Batjes, 2014; Dixon et al., 1994; Kindler et al., 2011). Moreover, elevated DOC concentrations can cause problems for drinking water treatment via

chlorination as DOC acts as a precursor of trihalomethanes, which have potentially carcinogenic and mutagenic properties (Alarcon-Herrera et al., 1994; Ledesma et al., 2012; Sadiq and Rodriguez, 2004). DOC can form complexes with organic pollutants (Hope et al., 1994) and toxic metals such as mercury (Ravichandran, 2004) or lead (Dörr and Münnich, 1991) and thus strongly influence drinking water quality. Several hypotheses have been proposed in order to explain the increase of DOC in many surface waters, including a decline in atmospheric sulphur deposition (Evans et al., 2006; Monteith et al., 2007;

Ledesma et al., 2016), a decline in nitrogen deposition (Musolff et al., 2016), reductions in ionic strength (Hruška et al., 2009), temperature increase (Freeman et al., 2001; Weyhenmeyer and Karlsson, 2009) and increased precipitation (Hongve et al., 2004). However, Roulet and Moore (2006) stress that it is difficult to isolate a single factor as there are many different variables influencing DOC production and export.

DOC export varies strongly between catchments. Annual exports between 1.2 and 56946 kg C $km^{-2}$ were found in a meta-

analysis of 550 catchments worldwide (Alvarez-Cobelas et al., 2012). Besides external factors influencing runoff, e.g. precipitation, a multitude of internal landscape-based factors may influence DOC export such as temperature controls on production (Moore et al., 2008; Wen et al., 2019) or chemical parameters such as pH and ionic strength (Hruška et al., 2009; Monteith et al., 2007) as well as redox processes (Knorr, 2013). Of particular relevance for DOC mobilization may be land cover type and changes in land use (Seybold et al., 2019; Larson et al., 2014; Aitkenhead-Peterson et al., 2007) as mobilization

processes depend on the DOC source area. Wetlands and the riparian zone are often particularly important DOC sources to streams and lakes as they are often characterized by large amounts of stored carbon, including DOC, which is easily mobilized (Harrison et al., 2005; Creed et al., 2008; Ogawa et al., 2006; Zarnetske et al., 2018; Weiler and McDonnell, 2006; Ledesma et al., 2015; Musolff et al., 2018).



Many studies have shown that DOC concentrations usually increase with discharge (Q) (Hobbie and Likens, 1973; McDowell
and Fisher, 1976; Meyer and Tate, 1983; Easthouse et al., 1992). However, at the event-scale, this concentration-discharge
relationship is seldom linear and hysteretic loops have been observed at many sites, the direction and shape of which being a
useful indicator for the underlying mobilization mechanisms. Clockwise hysteretic loops are generally explained by a DOC
source close and well connected to the stream (Blaen et al., 2017; Hood et al., 2006; Vaughan et al., 2017), flushing of DOC
from upper soil horizons during the rising limb (Buffam et al., 2001; Jeong et al., 2012) or a depletion of the DOC source
during the course of an event (Bowes et al., 2009; House and Warwick, 1998; Jeong et al., 2012). Anti-clockwise hysteretic
loops usually indicate a delayed activation of DOC source areas being located further away from the stream (Hood et al., 2006;
Vaughan et al., 2017) but also in terms of longer transit times (Musolff et al., 2017) or due to the change of flow pathways
(Brown et al., 1999; Hagedorn et al., 2000; Schwarze and Beudert, 2009; Strohmeier et al., 2013; Cerro et al., 2014) along
with changes in the connectivity (Ågren et al., 2008; Pacific et al., 2010).

Hydrologic connectivity is generally regarded to be of paramount importance for biogeochemical fluxes through watersheds.
As it controls runoff response during events, it has a direct impact on solute export (Kiewiet et al., 2020; Detty and McGuire,
2010). Therefore, it influences nutrient dynamics across different temporal and spatial scales (Covino, 2017). Hydrological
connectivity and therefore runoff and solute response are dependent on the antecedent hydrological conditions in a catchment
and the event characteristics itself (Penna et al., 2015; Detty and McGuire, 2010; McGuire and McDonnell, 2010) and appears
to be controlled by watershed topography and morphology. Weiler and McDonnell (2006) showed that the hillslope shape
influences the hysteresis pattern of DOC during storm events as mobilization processes differ with the geometrical properties
of hillslopes. Watershed morphology appeared to strongly influence C transport and storage in streams as the temperature
sensitivity of respiration depends on geomorphic features (Jankowski and Schindler, 2019).

In this study, we investigated DOC mobilization during different rain events and hydrological conditions at two different
topographical positions of a headwater stream within the Bavarian Forest National Park (BFNP) using high-resolution in-
stream DOC spectrometers. In particular, we compare two different parts of the catchment with regards to DOC-Q-
relationships and DOC export during four precipitation events in order to understand the influence of event size, antecedent
hydrological conditions and topography on DOC mobilization and export. We hypothesize that DOC mobilization processes
and DOC export are controlled by the underlying hydrological processes, in particular connectivity, and are strongly affected
by the antecedent hydrological conditions and event size. We further hypothesize that different topographical positions (steep
hillslopes vs. flat riparian zones) within the catchment promote different hydrological mobilization and transport processes
and therefore, DOC-Q relationships and DOC export will differ between the two parts of the catchment.





## 2 Material and Methods

### 2.1 Study site

The Große Ohe catchment (19.2 km$^2$) is part of the BFNP, which is located in Southeast Germany and shares a border with the Czech Republic. The BFNP covers an area of 243 km$^2$. Measurements were conducted in the nested sub-catchments Hinterer Schachtenbach (3.5 km$^2$) and Markungsgraben (1.1 km$^2$), which are part of the Große Ohe catchment, an experimental forested catchment run by the BFNP (see Table 1).


**Table 1 Characteristics of the catchment Große Ohe and the studied sub-catchments Markungsgraben (MG) and Hinterer Schachtenbach (HS)**

| Catchment | Große Ohe | Hinterer Schachtenbach (HS) | Markungsgraben (MG) |
|---|---|---|---|
| **Area (km$^2$)** | 19.2 | 1.5<br><br>3.5 (including Markungsgraben and Kaltenbrunner Seige) | 1.1 |
| **Elevation (m.a.s.l.)** | 770 - 1447 | 771 - 1085 | 888 - 1355 |
| **Mean slope (°)** | 7.7 | 7.4 | 15.9 |
| **Soils (%)** | | | |
| Cambisols | 64 | 66 | 55 |
| Podzols | 11 | 0 | 34 |
| Hydromorphic soils | 21 | 34 | 5 |
| Lithic Leptosol | 4 | 0 | 6 |
| **Vegetation (%)** | | | |
| Rejuvenation | 31 | 21 | 57 |
| Deciduous forest | 43 | 42 | 29 |
| Coniferous forest | 7 | 17 | 4 |
| Mixed forest | 18 | 19 | 8 |
| Other | 1 | 1 | 2 |

The Hinterer Schachtenbach catchment includes the sub-catchments Markungsgraben and Kaltenbrunner Seige (Fig. 1). Elevation in the catchment Große Ohe ranges from 770 to 1435 m. a. s. l. with a mean slope of 7.7°, whereas Markungsgraben represents the upper part of the catchment with a steeper mean slope (15.9°) than Hinterer Schachtenbach (7.4°). The geology

of the Große Ohe catchment is dominated by biotite granite and cordierite-sillimanite gneiss. The soils are mainly cambisols, podzols and hydromorphic soils, whereby the proportion differs between the sub-catchments. The mean annual precipitation

(1990 – 2010) is 1600 mm in the Markungsgraben sub-catchment and 1379 mm in the Hinterer Schachtenbach sub-catchment. Mean annual temperature is 6.2 °C at a station 3.8 km east of the catchment outlet. The entire catchment is almost exclusively covered by forest. However, large parts of the catchment are in a stage of rejuvenation due to bark beetle outbreaks in the mid-1990s and 2000s (Beudert et al., 2015). Dominant tree species in the forest are Norway spruce (*Picea abies,* 70 %) and European beech (*Fagus sylvatica*) (Table 1).

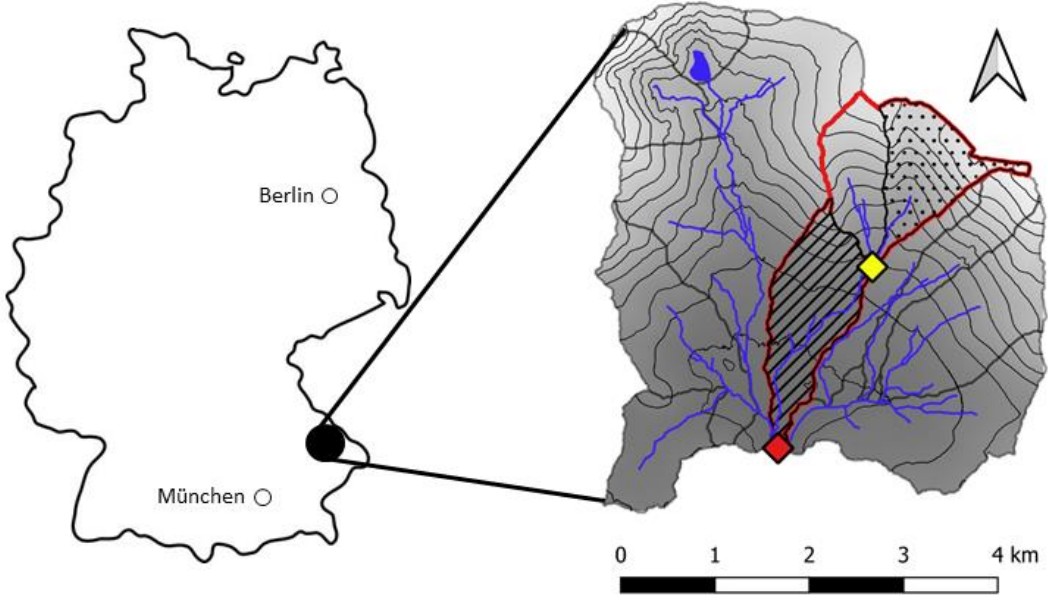


**Figure 1: The Große Ohe catchment is located in South-East Germany (left side). The continuous sampling sites were located at the outlet of the Markungsgraben (yellow diamond) and Hinterer Schachtenbach (red diamond). The red outline shows the total area contributing to the outlet of Hinterer Schachtenbach, including the sub-catchment Markungsgraben (dotted), Hinterer Schachtenbach (hatched) and Kaltenbrunner Seige (no pattern).**


## 2.2. Collection of field data

### 2.2.1 Location of the two continuous sampling points at different topographical positions

Two continuous sampling points with contrasting topographical positions were established for the period from June 2018 until October 2020 (Fig. 1): MG was located next to the gauging station of Markungsgraben (888 m.a.s.l.) in the steeper part of the

Große Ohe catchment, slightly upstream of the confluence of Markungsgraben and Kaltenbrunner Seige. As part of the Große Ohe hydrological monitoring discharge has been continuously measured at this station since 1988. HS (771 m.a.s.l.) was located shortly before the outlet of the Hinterer Schachtenbach into the Seebach and subsequently into the Große Ohe river. In





the following, the acronyms MG and HS stand for the respective measurement sites. We refer to the subcatchment Markungsgraben as the "upper catchment" and to the subcatchment Hinterer Schachtenbach as the "lower catchment". The total catchment includes the subcatchment Kaltenbrunner Seige that was not part of the monitoring presented here.

### 2.2.2 Climate and groundwater data

Precipitation and air temperature data were provided by the BFNP. Precipitation has been measured since 1978 at the station Taferlruck (N 48° 56.182 E 13° 24.819), close to our sampling site HS (Fig. 1, red diamond) and at the station Racheldiensthütte (N 48° 57.309 E 13° 25.544), next to our sampling site MG (Fig. 1, yellow diamond). In order to compare monthly precipitation amounts during the sampling period 2018, 2019 and 2020, long-term mean values for the two stations for the period from 1990 to 2010 were calculated. As temperature data were not available for both stations during the same period, data measured at Waldhäuser (N 48° 55.771 E 13° 27.890, 953 m.a.s.l., not visible in Fig. 1), a station 3.8 km east of the catchment outlet, was used. Groundwater level data was provided by the Bavarian State Office for Environment for Wilde Rast, which is located uphill from MG (964 m.a.s.l.). For the other two locations, Eschenhäng and Schachtenebene, groundwater level data was provided by the BFNP. Eschenhäng (969 m.a.s.l.) is located uphill from MG and Schachtenebene (819 m.a.s.l.) is located close to HS.

### 2.2.3 Discharge measurements

Starting in June 2018, the water level was measured every 15 minutes at HS using a pressure transducer (Solinst Canada Ltd., Georgetown, Canada and SEBA Hydrometrie GmbH, Kaufbeuren, Germany). Flow velocities were measured periodically at the same location with an electromagnetic current meter (FlowSens, SEBA Hydrometrie GmbH, Kaufbeuren, Germany) and via tracer-dilution (TQ-S, Sommer Messtechnik, Koblach, Austria). Corresponding discharge was calculated following Kreps (1975). For MG, the discharge data for the complete sampling period were taken from the data base of the Bavarian State Office for Environment (2020). Discharge values of MG were interpolated between the measured stepwise discharge changes, which were due to a low resolution of discharge measurements.

### 2.2.4 Continuous measurements of DOC concentration at different topographical positions

DOC concentrations were measured continuously in-situ at MG and HS using two UV-Vis spectrophotometers (spectro::lyser, s::can Messtechnik GmbH, Vienna, Austria). The spectrometric devices recorded the absorption spectrum of stream water from 200 to 750 nm with a resolution of 2.5 nm every 15 minutes. DOC concentrations were quantified using the internal calibration based on the absorption values using the software ana::pro. The DOC concentrations measured by the UV-Vis spectrophotometers were calibrated using 21 (MG) and 52 (HS) grab stream samples taken over the course of the sampling period at various discharge conditions. Samples were filtered in the field using polyethersulphone membrane filters (0.45 µm) during the first sampling campaigns and polycarbonate track etched membrane (0.45 µm) starting in October 2018. All samples were stored until further analysis at 4°C. DOC concentrations of the grab samples were analyzed in the laboratory by thermo-





catalytic oxidation (TOC-L-Analyzer, Shimadzu, Kyoto, Japan). For further analysis the calibrated values ($R^2$ for HS = 0.98,

$R^2$ for MG = 0.77) were used. As no drift of the DOC concentration could be identified in the measured signal, we decided against a correction for biofouling as done by Werner et al. (2019). However, the sensor optics were manually cleaned in the field every two weeks using cotton swabs. DOC concentrations were measured from June to November 2018, from April to November 2019 and from April to October 2020. In this study, we focus on the analysis of four large events, which were characterized by contrasting antecedent hydrological conditions. The four events took place in June 2018, October 2018, May

2019 and September 2020.

## 2.2 Analysis of event characteristics

The last time step measuring baseflow was classified as the start of an event. The start of Q increase was defined as the first value higher than the sum of the average Q and the twofold standard deviation of the three hours prior to the start of

precipitation. The end of an event was defined as the moment when Q returned to pre-event values again. In October 2018, the observed event was quickly followed by a second event, which led to slightly overlapping hydrographs at HS. As Q was already within 0.003 $m^3 s^{-1}$ of pre-event baseflow, we did not simulate the end of the first hydrograph but rather used the incomplete hysteretic loop for load calculations. In September 2020, post-event baseflow was higher than pre-event baseflow and was defined as the first value lower than the three hour average plus the twofold standard deviation prior to the start of the

next event several days later.

For each event DOC concentrations were plotted as a function of Q and the hysteresis index ($h$) as proposed by Zuecco et al. (2016) was calculated. The index is based on the computation of definite integrals at fixed intervals of normalized Q and represents the difference between the integrals on the rising and falling curves computed for the same interval. A positive hysteresis index indicates a clockwise hysteresis, whereas a negative hysteresis index indicates an anti-clockwise hysteresis.

The larger the absolute value of $h$, the wider is the hysteretic loop. Additionally, DOC fluxes were calculated by multiplying the 15 min discharge value with the corresponding 15 min DOC concentration. These were cumulated to a total DOC load per event.

In order to compare the specific characteristics of the precipitation events and the response of discharge and DOC concentrations in the stream, we used the following parameters. $P_{tot}$ is the total amount of precipitation during the event. The

antecedent precipitation ($AP_{14}$) is the cumulative precipitation 14 days prior to the start of the event following van Verseveld et al. (2009). The sum of $P_{tot}$ and $AP_{14}$ represents the total catchment wetness. The P-Q lag is defined as the time (in minutes) between the start of precipitation to the start of the first Q increase as a response to the precipitation. The Q lag time is the time in minutes from the first Q increase to the Q peak ($Q_{max}$). DOC lag time is the time in minutes from $Q_{max}$ to the DOC peak ($DOC_{max}$). We also introduce a new index called $DOC_{90}$. It represents the period of time in minutes during which the DOC

concentrations exceed 90% of $DOC_{max}$ during the event. It provides a measure for the formation of a plateau of the hysteretic





loop before the decrease of DOC concentrations. The precipitation specific DOC load is defined as the DOC load per precipitation amount (kg km$^{-2}$ mm$^{-1}$).

## 3 Results

### 3.1 Event characteristics and discharge dynamics

The annual precipitation at HS was 1126 mm in 2018, 1125 mm in 2019 and 1072 mm in 2020, which was considerably lower than the long-term average of 1379 mm (1990 and 2010). At MG, annual precipitation was 1274 mm in 2018, 1380 mm in 2019 and 1293 mm in 2020 compared to 1600 mm (1990 – 2010). The annual mean temperature was between 1.4 and 1.6 °C higher than the long-term average of 6.2°C. The sampling periods in all three years were overall characterized by warmer temperatures and less precipitation when compared to the long-term mean monthly precipitation sums and monthly average

temperatures (Fig. 2).

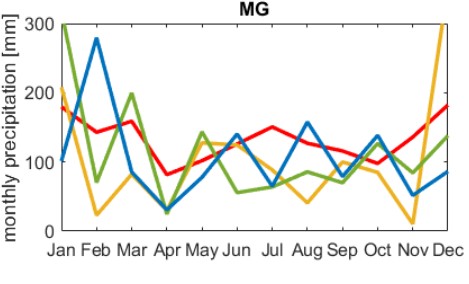

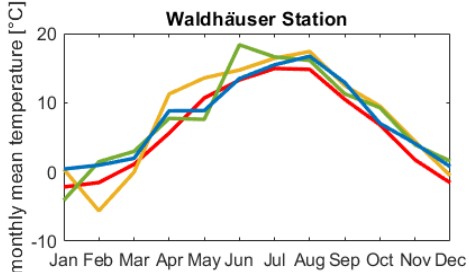

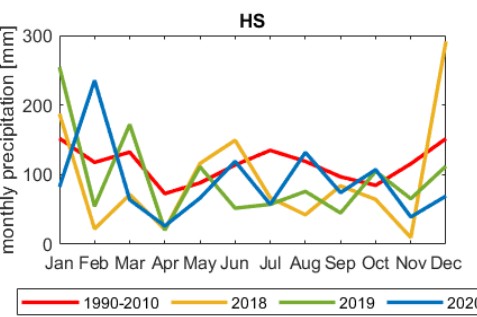

**Figure 2: Monthly precipitation sums (left) at the upper site (MG) and the lower site (HS) and monthly mean temperatures (right) in 2018, 2019 and 2020 in comparison to the long-term mean of the years 1990 - 2010.**

In general, the two events in late spring/early summer (June 2018 and May 2019) were preceded by long periods of higher

groundwater tables following winter precipitation and snowmelt (Fig. 3). The two early fall events (October 2018 and September 2020) followed dry summers, characterized by pronounced groundwater table declines. The three events analyzed in June 2018, May 2019 and October 2018 were of similar size but differed markedly in their antecedent wetness conditions (Table 2). In June 2018, several larger events occurred prior to the analyzed event, which led to a very high AP$_{14}$ at both sites. The value at HS exceeded the value at MG due to a large local event in the beginning of June. In May 2019, a succession of



several events led to an intermediate $AP_{14}$, compared to event in June 2018. The amount of precipitation was also slightly lower in May 2019 than during the June event in 2018. The October event followed a prolonged dry period leading to the lowest $AP_{14}$ value of the four events. The event in September 2020 was much larger than the other three events with almost twice the amount of precipitation. The $AP_{14}$, however, was very low and comparable to the October 2018 event.

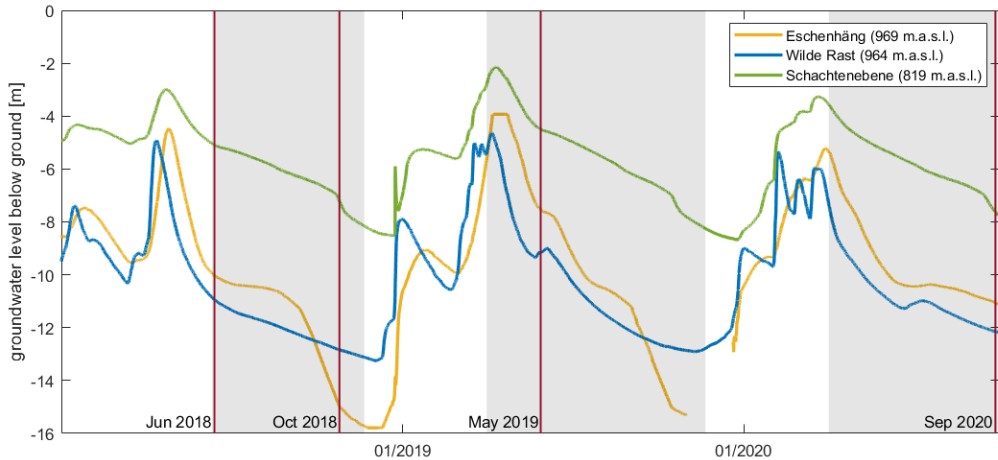

**Figure 3: Groundwater levels at three locations from 2018 to 2020. Eschenhäng and Wilde Rast are located in the sub-catchment Markungsgraben, Schachtenebene in the sub-catchment Hinterer Schachtenbach. The shaded areas represent the three sampling periods and the red lines the selected events.**

This offered the opportunity to evaluate the effect of event size, representing the total amount of precipitation during an event, using these two early fall events with similar $AP_{14}$. In contrast, total catchment wetness values ($AP_{14}$ + $P_{tot}$) of the events in

May 2019 and September 2020 were very similar, providing an additional opportunity to evaluate the relative importance of event size vs. antecedent hydrological conditions. The event in June 2018 led to the largest $Q_{max}$ of the four events at HS. At MG, the highest $Q_{max}$ value for the four studied events was measured for the October 2018 event (Fig. 4a), which was still not close to mean high-flow discharge of MG of 0.869 $m^3\,s^{-1}$ nor to a HQ1 event (0.7 $m^3\,s^{-1}$). Baseflow of MG during the sampling period varied mostly between the lowest low-flow discharge (0.006 $m^3\,s^{-1}$) and the mean low-flow discharge (0.013 $m^3\,s^{-1}$).

For HS no comparison is possible, as discharge monitoring only started in June 2018. $Q_{max}$ values at HS varied as a function of total catchment wetness, whereas at MG, $Q_{max}$ was not significantly affected by total catchment wetness. MG generally showed a faster Q response than HS represented by the shorter P-Q lag time. During the events in June 2018 and May 2019, the P-Q lag time was very similar at both locations. During the events in October 2018 and September 2020, following a dry summer, the P-Q lag time at HS was much longer than at MG. Q lag time at HS, however, was shorter than at MG during these

events and longer during the events in June 2018 and May 2019.



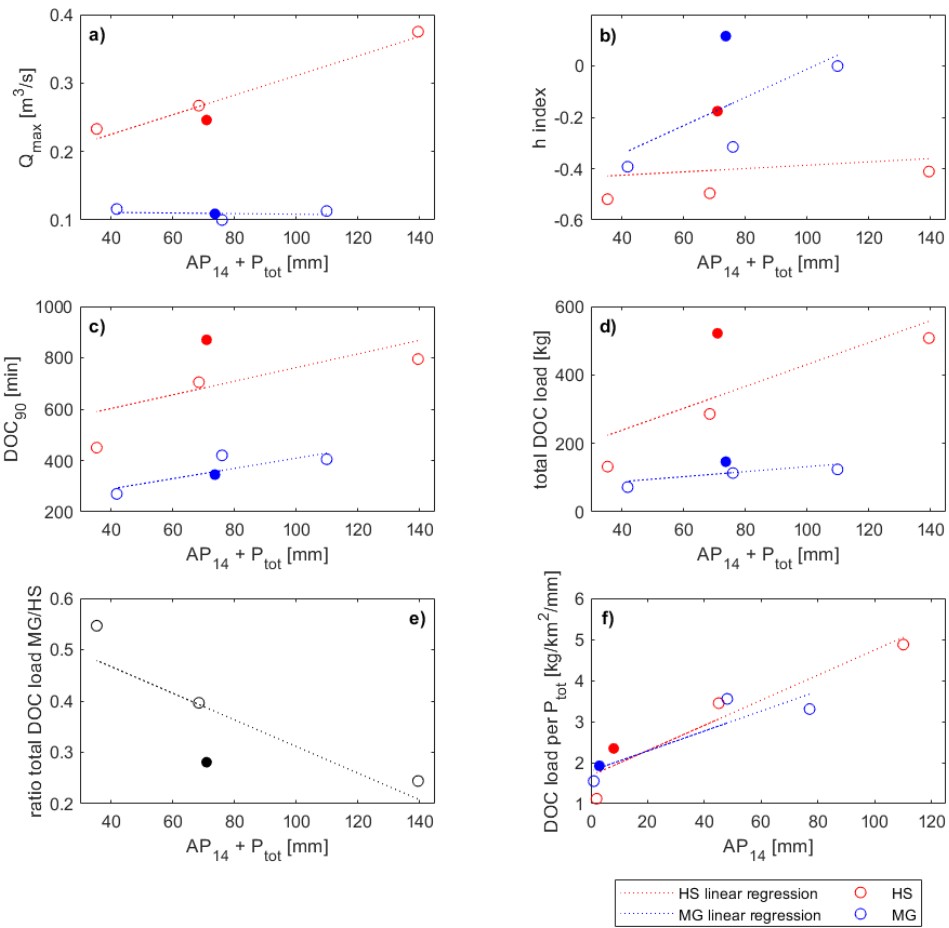

**Figure 4: a) – d) Event parameters as a function of the sum of $AP_{14}$ and $P_{tot}$ (total catchment wetness) at the lower site (MG) and the upper site (HS). The closed symbols indicate the September 2020 event with its much larger amount of precipitation. e) The ratio of the total DOC load of MG/HS as a function of total catchment wetness. f) Precipitation specific DOC load [kg km$^{-2}$ mm$^{-1}$] as a function of $AP_{14}$.**

### 3.2 DOC dynamics and DOC-Q-hysteresis patterns

Our long-term measurements showed that DOC concentrations during baseflow were very similar at the two locations and usually varied between 2 and 3 mg L$^{-1}$. At higher baseflow during wetter periods (June 2018 and May 2019), DOC concentrations were slightly higher than during dry periods (October 2018 and September 2020). In general, we observed rapidly rising DOC concentrations with rising discharge after precipitation events (Fig. 5). Peak DOC concentrations for the four studied events varied for HS between 10.2 – 18.6 mg L$^{-1}$ and for MG between 8.5 - 16.9 mg L$^{-1}$, without a clear relation to $P_{tot}$, $AP_{14}$ or total catchment wetness.



**Figure 5: DOC concentrations and Q during June 2018 (top row), October 2018 (second row), May 2019 (third row) and September 2020 (bottom row) at HS (left) and MG (right). The shaded areas highlight the events described in detail in Table 2. Note that the scaling of the y-axis for Q differs between HS and MG.**





**Table 2: Event characteristics of the four selected events at HS and MG: $P_{tot}$ (total precipitation in mm during the event), $AP_{14}$ (antecedent precipitation, cumulative precipitation in mm 14 days prior to event start), total catchment wetness (sum of $P_{tot}$ and $AP_{14}$ in mm), P-Q lag time (time in minutes from the start of precipitation to the start of Q increase), Q lag time (time in minutes from the start of Q increase to $Q_{max}$), h (Hysteresis Index following Zuecco et al., see section 2.2), DOC lag time (time in minutes from $Q_{max}$ to $DOC_{max}$), maximum DOC concentration ($DOC_{max}$ in mg $L^{-1}$), $DOC_{90}$ (time in minutes during which the DOC concentrations exceed the 90% of $DOC_{max}$), absolute DOC load in kg, and precipitation specific DOC load (kg km$^{-2}$ mm$^{-1}$).**

|  | Start Date | $P_{tot}$ (mm) | $AP_{14}$ (mm) | Total catchment wetness (mm) | P-Q lag time (min) | Q lag time (min) | $Q_{max}$ (m$^3$ s$^{-1}$) |
|---|---|---|---|---|---|---|---|
| HS | 12.06.2018 | 30 | 110 | 140 | 60 | 345 | 0.375 |
|  | 23.10.2018 | 33 | 2 | 35 | 1110 | 270 | 0.233 |
|  | 27.05.2019 | 24 | 45 | 69 | 30 | 540 | 0.267 |
|  | 25.09.2020 | 63 | 8 | 71 | 225 | 1755 | 0.252 |
| MG | 12.06.2018 | 33 | 77 | 110 | 30 | 225 | 0.113 |
|  | 23.10.2018 | 41 | 1 | 42 | 15 | 1170 | 0.116 |
|  | 27.05.2019 | 28 | 48 | 76 | 30 | 375 | 0.100 |
|  | 25.09.2020 | 71 | 3 | 74 | 15 | 2040 | 0.109 |

|  | Start Date | $h$ | DOC lag time (min) | $DOC_{max}$ (mg $L^{-1}$) | $DOC_{90}$ (min) | DOC load (kg) | Precipitation specific DOC load (kg km$^{-2}$ mm$^{-1}$) |
|---|---|---|---|---|---|---|---|
| HS | 12.06.2018 | -0.411 | 165 | 17.1 | 795 | 508 | 4.9 |
|  | 23.10.2018 | -0.519 | 255 | 12.8 | 450 | 132 | 1.1 |
|  | 27.05.2019 | -0.496 | 465 | 10.2 | 705 | 286 | 3.4 |
|  | 25.09.2020 | -0.176 | 285 | 18.6 | 870 | 522 | 2.3 |
| MG | 12.06.2018 | 0.000 | 120 | 16.9 | 405 | 124 | 3.3 |
|  | 23.10.2018 | -0.393 | 90 | 15.2 | 270 | 72 | 1.5 |
|  | 27.05.2019 | -0.315 | 225 | 8.5 | 420 | 113 | 3.6 |
|  | 25.09.2020 | 0.116 | 0 | 14.6 | 345 | 146 | 1.9 |





The increase of DOC concentrations occurred faster at MG than at HS, which led to shorter DOC lag times at MG (Table 2). The longest DOC lag time was found in May 2019. However, the event with the shortest lag time was in September 2020 at

MG, whereas at HS it was in June 2018. DOC concentrations at HS generally maintained their $DOC_{90}$ concentrations during a longer time than at MG and values were higher at HS during all events (Fig. 4c). This resulted in wider hysteretic loops at HS than at MG (larger absolute values of $h$) where concentrations decreased soon after reaching the DOC peak.

The DOC-Q-relationships during events showed almost exclusively anti-clockwise hysteresis patterns at both sites (Fig. 6) resulting in negative hysteresis indices ($h$, see Table 2). Exceptions were the events in June 2018 and September 2020 at MG,

where the hysteresis exhibited no clear loop form and an $h$ value close to zero. With increasing total catchment wetness, the $h$ value approached zero at MG (Fig. 4b). At both sites, the lowest absolute $h$ value, and thus the narrowest hysteretic loop, was found in September 2020 during the largest event. The largest absolute $h$ value, and thus the widest hysteretic loop, was found in October 2018 following the extreme dry period.





**Figure 6: DOC-Q-Hysteresis during events in June 2018, October 2018, May 2019 and September 2020 at HS and MG. The dots represent 15-minute time steps starting from dark blue to green and represent the events that are marked by the grey areas in Fig. 5.**





### 3.3 DOC export from sub-catchments during different hydrological conditions

The absolute DOC export was higher during all events at HS than at MG due to the larger catchment area. We observed the highest DOC export at both locations during the event September 2020 with 522 kg at HS and 146 kg at MG (Table 2). At both locations, we observed the lowest DOC export during the October 2018 event with 132 kg and 72 kg, respectively. At HS, the total DOC load increased with total catchment wetness (Fig. 4d). At MG, this relationship was not as pronounced. However, at both sites, the precipitation specific DOC load increased clearly with the $AP_{14}$ (Fig. 4f).

Assuming that the difference between the total DOC arriving at HS and the total DOC arriving at MG originated from the sub-catchments KS and HS, the relative contribution of MG to DOC export can be evaluated. The comparison of the ratio of cumulative water volumes and of cumulative DOC loads measured at MG and HS (ratio MG/HS in Fig. 7) illustrated the varying relative contribution of the upper catchment. Although it is unlikely that the entire catchment area served as a DOC source area, we compared the contribution of the upper catchment to the expected ratio based on catchment area alone, which

would be 0.32. However, only the events in June and May showed MG/HS ratios close to this value for both cumulative Q and DOC. The events in October and September showed a considerably higher MG contribution for both Q and DOC, especially at the beginning of the events. Over time, the ratios approached the expected value of 0.32 as the contribution of MG decreased. In June and May, the Q and DOC ratios showed similar values with Q being slightly higher than DOC. In contrast, during October and September, DOC ratios were higher than Q ratios most of the time. The MG contribution to the total DOC load

at HS was strongly negatively correlated to the total catchment wetness of the catchment (Fig. 4e).





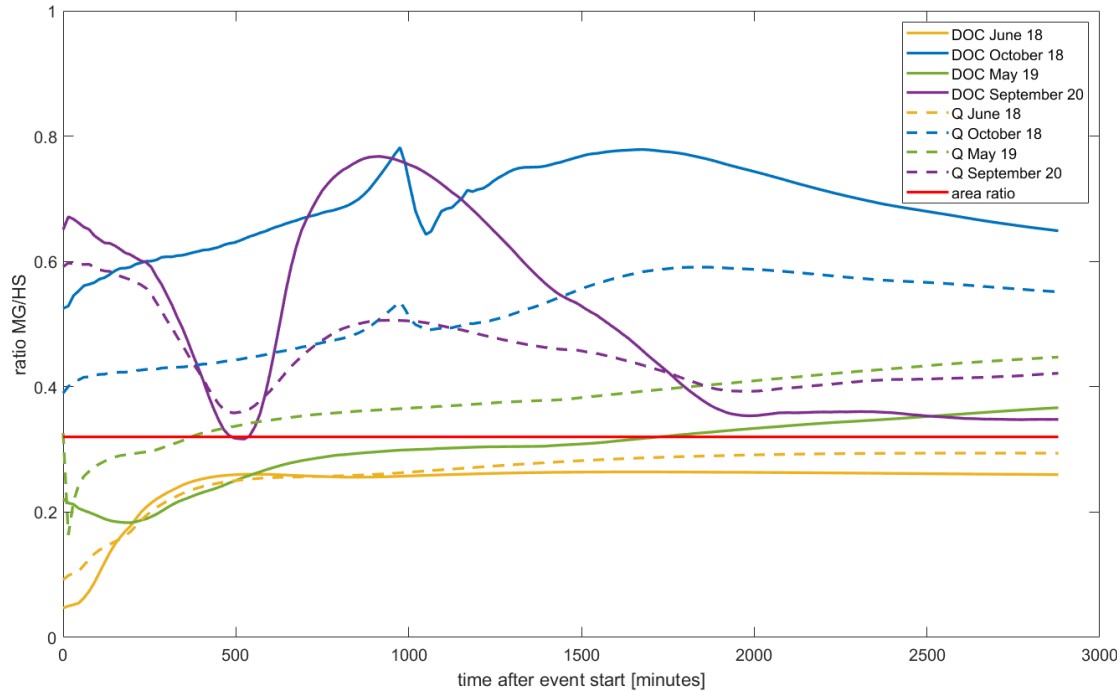

**Figure 7: Ratio of cumulative Q (dashed lines) and DOC load (solid lines) at MG and HS during the four selected events. The red line indicates the expected ratio in terms of area (0.32). We included a three hour lag of the arrival from MG at HS based on flow velocity measurements taken at high discharge.**

## 295 **4 Discussion**

### 4.1 Discharge response depends on topographical positions and antecedent hydrological conditions

The sampling period was defined by prolonged dry conditions and few rain events. However, we were able to compare the reaction of the catchment at two contrasting locations to three similarly sized events and to one event that was unusually large for the observation period. The events were characterized by contrasting antecedent hydrological conditions.

In contrast to MG, $Q_{max}$ at HS was clearly linked to the total catchment wetness prior to the event (Fig. 4a). This correlation suggests that precipitation events are not the only driver of Q generation but that Q generation in the lower catchment depends highly on the antecedent hydrological conditions. The Q lag times were longer at HS than at MG during the events with wet antecedent conditions and shorter during the events with dry antecedent conditions. Even more pronounced, P-Q lag times at HS exceeded the values at MG especially after the prolonged dry periods indicating a slower response to rainfall events at the

lower catchment. To some extent, this observation can be attributed to the larger catchment area contributing to water fluxes at HS. As water arrives from further away, it causes a delayed Q increase. We argue that topographic differences also play a key role in the response of discharge to rainfall events, especially under dry conditions. Steeper slopes in the upper catchment





generate a faster delivery of water to the stream than at the lower catchment, which is characterized by a low gradient topography. Processes such as the transmissivity feedback (Bishop et al., 2004), which describes how a rising groundwater

table together with increasing hydraulic conductivity in upper soil layers may lead to a delayed increase in discharge in the receiving stream (Frei et al., 2010), could be of importance in the extensive riparian zone of the lower catchment. We suggest that hydrological connectivity in the lower catchment is the major driver for delivering water to the stream, being dependent on topography as well as on the antecedent hydrological conditions (Detty and McGuire, 2010; Penna et al., 2015; McGuire and McDonnell, 2010).

As seen in Figure , the upper catchment delivered relatively more water than the lower catchment after long dry periods. This proportion changed over time with a larger fraction of discharge being contributed by the lower catchment. The relationship between $Q_{max}$ and total wetness suggests that the lower catchment is highly dependent on the establishment of hydrological connectivity for discharge generation, in contrast to the upper catchment. The lower catchment with the low topography and large riparian zone has a larger water storage capacity in contrast to the upper catchment due to deeper soils.  After dry periods,

the soils in the lower catchment need to be rewetted before flow pathways towards the stream become activated, whereas in the steeper upper catchment runoff generation starts sooner and thus contributes relatively more to the total flux, especially at the beginning of the event. Over time, the water contribution becomes balanced as flow paths in the lower catchment are connected to the stream. Zimmer and McGlynn (2018) observed that after wet antecedent conditions, the lowlands contributed more to runoff, whereas after dry antecedent conditions, the headwaters became more important. The correlation between the

total catchment wetness and $Q_{max}$ also shows that the antecedent conditions are pivotal for Q generation in the lower catchment as they strongly influence top and subsoil water stores. Even during the large event in September 2020, $Q_{max}$ of HS was lower than during the events in May 2019 and June 2018, which were characterized by a smaller precipitation amount but much wetter antecedent conditions. Event size alone is therefore not a good predictor for peak discharge values.

### 4.2 The interplay of event size and antecedent hydrological conditions as a controlling factor for DOC mobilization and
export

As observed in previous studies, DOC concentrations did not vary strongly during baseflow but increased rapidly in response to increasing discharge following precipitation events. Concentrations were generally in the expected range for a forested catchment in low mountain ranges (Musolff et al., 2018; Schwarze and Beudert, 2009). Larger events generally lead to higher DOC concentrations in streams. In some catchments, successive events can lead to the depletion of DOC sources (Bartsch et

al., 2013; Butturini et al., 2006; Jeong et al., 2012; van Verseveld et al., 2009) and consequently to lower DOC concentrations. Our results showed that after successive events, peak DOC concentrations did not decrease (Fig. 5, May 2019). We conclude therefore that DOC fluxes in the stream are limited by transport from the riparian zone in the Große Ohe catchment rather than source limited, at least for the event frequency and magnitude observed during the study period.  This is also the case for most of the >1000 catchments studied by Zarnetske et al. (2018), which varied in area, stream order and ecoregion type. Our data

show that transport limitation is clearly linked to the antecedent hydrological conditions, which strongly influence the



hydrological connectivity of the catchment as it has been shown in other studies as well (Detty and McGuire, 2010; Penna et al., 2015).

In the following, we will focus on processes referring to the entire catchment, which could be observed at HS. $AP_{14}$ as well as total catchment wetness and total DOC load were positively related (Fig. 4d). The event in June 2018 with the highest $AP_{14}$

and highest total catchment wetness led to the highest DOC concentrations and the highest DOC load of all similarly sized events. This becomes particularly clear in comparison with the slightly larger event in the following October. The strong relation between $AP_{14}$ and the precipitation specific DOC load confirms the importance of antecedent wetness conditions for DOC mobilization during events (Fig. 4f). The link between wet antecedent conditions, a higher connectivity and subsequently a higher DOC export was also observed by Zimmer and McGlynn (2018) in a small catchment in North Carolina. This

observation is similar to Inamdar and Mitchell (2006) who suggested that with increasing soil moisture previously disconnected DOC source areas become active leading to a strong increase in DOC export. We observed hysteretic loops generally showing a slow DOC mobilization with discharge increase in the riparian zone. However, the $h$ index was approaching zero with increasing total catchment wetness indicating faster mobilization processes (Fig. 4b). We also observed a clear link between total catchment wetness and $DOC_{90}$ (Fig. 4c). The persistently high concentrations in combination with a high discharge

generation due to the existing connectivity could then cause the high DOC export during events following wet antecedent conditions.

We propose that the delayed DOC increase observable in the stream upon an increase of discharge (Figure 6) reflects not only the larger catchment size but also the slow saturation of DOC-rich soil layers and the slow establishment of connectivity to the stream, similar to systems where the transmissivity feedback mechanisms is the dominant control on DOC fluxes (Bishop et

al., 2004). In addition, connectivity with near-stream small pools could contribute to a delayed increase of DOC concentrations. We observed pools in the riparian zone that only contained water during very wet conditions. These pools then connected to a small tributary of the stream. Analyses of the pool water showed very high DOC concentrations, between 30 and 50 mg L$^{-1}$ (data not shown). The possibility that these pools contribute to the DOC export later during the event is also supported by the observation that topographic depressions can play a very important role in DOC accumulation and DOC transport to the stream

(Ploum et al., 2020). In general, the riparian zone saturates over the course of a precipitation event (Ledesma et al., 2015; Tunaley et al., 2016). This process happens faster after wet antecedent conditions, which lead to a larger hydrologically connected area of the riparian zone with longer flow paths in organic-rich layers on the way to the stream. These lateral subsurface flows through the upper soil horizon are an important process for DOC mobilization (Bartsch et al., 2013; Birkel et al., 2017). If the soils are wet prior to an event, connected flow paths can quickly be established and DOC transport to the

stream occurs faster than during dry conditions. This could explain why the highest DOC exports were found during wet conditions, which is in contrast to other studies that have shown that DOC concentrations and export are especially high during events following longer dry periods due to stagnant pools in the stream (Inamdar and Mitchell, 2006; Granados et al., 2020) and an increased DOC production through oxidation of organic matter and accumulation after a dry and warm summer (Tunaley et al., 2016; Wen et al., 2019; Strohmeier et al., 2013). However, DOC export during the events in October 2018





and September 2020, following the dry summer, was relatively low due to the low discharge generation (see 4.1.) further supporting our hypothesis of transport limited DOC export.

Hydrological connectivity is not solely influenced by the antecedent hydrological conditions and storage capacities, but also by the event size (Correa et al., 2019; McGuire and McDonnell, 2010) as large events lead to the expansion of saturated areas (Tetzlaff et al., 2014). Several studies have shown that event size also plays an important role in DOC mobilization as the

largest DOC export of a catchment occurs during precipitation events. Large single events can contribute significantly to the annual DOC export in small catchments, whereby the event size is often more important than the event frequency (Raymond and Saiers, 2010; Raymond et al., 2016). The event in September 2020 was much larger than the other three events studied here.

Comparing the two early fall events which were characterized by equally dry antecedent conditions but different amounts of

precipitation (October 2018 and September 2020), we see that the Q peak was similar at both events although the September event had almost twice the amount of precipitation of October 2018. The lack of connectivity appeared to inhibit discharge generation in September 2020. Nevertheless, DOC export was much higher in September 2020 due to the duration of the event. We therefore conclude that the event size does have an important effect on DOC export when the antecedent conditions are similar. The two events in May 2019 and September 2020 had a similar value of total wetness ($AP_{14}$ and $P_{tot}$), but differed

markedly in $P_{tot}$ and $AP_{14}$, which led to clear differences in some event characteristics. The September event, a large event after dry conditions, was characterized by higher $DOC_{90}$ and higher $DOC_{max}$ values. Higher DOC concentrations thus prevailed for a longer time, which could be due to a larger available DOC pool after the warm summer months (Strohmeier et al., 2013; Tunaley et al., 2016; Wen et al., 2019). The hysteretic loop was narrower in September indicating a faster DOC mobilization due to the precipitation amount. This fast mobilization in combination with long-lasting high DOC concentrations led to a

higher DOC load in September 2020 than in May 2019, a small event after wet conditions. However, as described above, missing connectivity likely inhibited DOC export, especially at the beginning of the event in September 2020, which could explain the rather small difference in DOC load in comparison to the event in June 2018. During this event, the small amount of precipitation was offset by a very high $AP_{14}$ leading to almost the same DOC load as in September 2020 with less than half of the precipitation.

Following these observations, we suggest a hierarchy of controlling factors with respect to their relevance for discharge and DOC release. For Q generation, the antecedent wetness conditions are important, whereas the event size plays a minor role. For DOC export, however, the event size was of importance as a larger event resulted in a markedly higher DOC mobilization as observed in September 2020. Analyzing only the similarly sized events (June 2018, October 2018, May 2019), total catchment wetness controlled DOC export.






### 4.3 Clear differences in DOC mobilization and export between topographical positions

Antecedent hydrological conditions are also linked to storage capacities and connectivity, which in turn are strongly correlated with topography. In contrast to HS, we could not see a correlation between $Q_{max}$ and the total catchment wetness at MG indicating different hydrological runoff processes, which were less dependent on hydrological connectivity (Fig. 4a). Moreover, we did not observe a relation between the *h* index and the total catchment wetness in contrast to the observations at HS (Fig. 4b). At MG, DOC lag times were generally shorter and hysteretic loops narrower indicating faster DOC mobilization than at HS. As discussed in section 4.1., we argue that the steeper slopes at MG facilitate a fast connectivity and transport of water and consequently DOC to the stream. The events in June 2018 and September 2020 led to the only hysteretic loops that were not clearly anti-clockwise. The very fast increase and decrease of DOC concentrations led to 'eight'-shaped, almost clockwise loops at MG. In September 2020, the large event size led to a fast delivery of DOC to the stream. In June 2018, the wet antecedent hydrological conditions facilitated a fast connection of close DOC sources to the stream with the start of the event. Correa et al. (2019) made a similar observation with anti-clockwise hysteresis patterns of several rare earth elements in the downstream catchment areas and clockwise hysteresis patterns in upstream parts of the catchment, which they attributed to faster responding end members. Once reaching the maximum, DOC concentrations at MG generally started to decrease quite fast, whereas at HS they remained elevated over several hours, as seen in the $DOC_{90}$ values.

Although Li et al. (2015) suggest that vegetation and the abundance of lakes and wetlands are more important for DOC export than topography alone, other studies have shown that topography can strongly influence DOC mobilization in some catchments. Both Musolff et al. (2018) and Ogawa et al. (2006) observed a correlation between the topographic wetness index, an indicator for potential soil wetness linking slope and upslope contributing area, and DOC concentrations. A flat area would therefore tend to export more DOC than a steep area due to a higher general wetness. Surprisingly, calculation of DOC export revealed that the lower catchment did not always dominate export. The total DOC export per event was similar to the DOC export observed by van Verseveld et al. (2009). These authors observed export rates varying from below 50 kg km$^{-2}$ per storm for events with slightly more than 40 mm precipitation to 440 kg km$^{-2}$ for an event with 200 mm precipitation. Export rates at MG varied between 63 and 128 kg km$^{-2}$ and at HS between 37 and 148 km$^{-2}$. The total DOC load arriving at HS during an event was, as expected, larger than the DOC load arriving at MG due to the larger catchment area, which not only includes the part below MG but also the neighboring sub-catchment KS. Taking into account the upstream catchment area of the sites, one would expect around 32 % of DOC arriving at HS coming from MG based on the assumption that both sites have the same source strength and no DOC is lost from the stream. We use the value of 32 % for the purpose of comparison and do not intend to imply that the entire catchment area is a DOC source area. Figure 7 shows that the contribution of the upper catchment is much larger than 32 % under certain conditions and even exceeds the contribution of the lower catchment. This relative dominance of DOC originating from the upper catchment was most prominent in October 2018 and September 2020 after the dry summer and decreased with the total catchment wetness (Fig. 4e). During the large event in September 2020, the contribution of each catchment component changed over time. At the beginning of the event, considerably more DOC was





mobilized from the upper catchment. Over time, however, the role of the upper catchment for DOC export decreased and the

contribution from the lower catchment increased. One aspect of a large contribution of DOC from the upper catchment might be higher precipitation than at the lower catchment, which could lead to an increased DOC mobilization. The amount of Q arriving from the upper catchment is proportionally higher; however, the DOC proportion is even higher. Consequently, not all of the mobilized DOC from the upper catchment can be explained by the higher discharge induced by a higher precipitation. These observations contrast with our expectations that the lower catchment would be more important for DOC export than the

upper catchment. Although the upper catchment area has high deadwood content that might lead to a large DOC pool (Schwarze and Beudert, 2009), the vegetation is currently dominated by mostly regenerating Norway spruce forests and by deciduous trees, whereas the lower catchment is partly covered by mature coniferous forest and riparian peatland. Previous studies have shown that soil layers beneath conifers are usually richer in DOC than beneath deciduous trees (Schwarze and Beudert, 2009; Borken et al., 2011), which could contribute to higher in-stream DOC concentrations. In addition, a higher

DOC production would be expected in the lower catchment due to the elevation differences and subsequently different temperatures (Borken et al., 2011; Tunaley et al., 2016; Wen et al., 2019; Andersson et al., 2000). Another reason for a higher DOC export from the lower catchment would be the importance of large riparian zones for DOC mobilization (Ploum et al., 2020; Mei et al., 2014; Ledesma et al., 2015; Ledesma et al., 2018; Inamdar and Mitchell, 2007; Strohmeier et al., 2013; Musolff et al., 2018). The importance of the riparian zone for in-stream DOC concentrations is confirmed by DOC

measurements performed in a low elevation sub-catchment next to HS with more than 40 % hydromorphic soils (Beudert et al., 2012) and with stream DOC concentrations being higher (6.3 mg L$^{-1}$) at baseflow than in the lower catchment.

Nevertheless, the upper catchment contributes strongly to the total DOC export of the catchment, especially after dry antecedent conditions We suggest that the main reason for this reversed contribution to total DOC export is rather decreased DOC export from the lower catchment during events after dry conditions than an increased DOC export of the upper catchment

during wet conditions. The variation of the total DOC load between the events was much higher at HS than at MG, which indicates a high dependency on the hydrological preconditions. As explained above, connectivity seems to be important for the DOC mobilization especially in the lower catchment, where the missing connectivity in October and September seems to inhibit DOC mobilization more than in the upper catchment. Another reason for the reduced contribution of the lower catchment could be that the riparian pools mentioned above are not connected and thus an important DOC source is not active.

We suggest that the different topography also plays a role as flow paths in the lower catchment are not as easily connected as in the steep area. Due to the low gradient terrain, saturated soils are necessary to transport water laterally through the DOC rich soil layers towards the stream. In contrast, the steeper hills in the upper catchment facilitate a rapid transport to the stream independent of the rather dry soil layers leading to a relatively high DOC export during the rain events. The change of contribution over the course of the large event in September 2020 confirms our reasoning that connectivity is the limiting

factor of DOC mobilization in the lower catchment.

## 5 Conclusions

In this study, we showed that DOC mobilization and export depend on event size, antecedent hydrological conditions and catchment topography. The amount of precipitation has a strong impact on the DOC export. However, if events are similar in size, the antecedent hydrological conditions control DOC export. After wet hydrological conditions, we observed a larger DOC

export than after dry conditions. Especially in the lower catchment, DOC was not mobilized at the beginning of an event as the soils in the riparian zone first needed to saturate. This led to a disproportionate contribution of the upper catchment to the total DOC export early in the events characterized by dry antecedent hydrological conditions. We cannot provide data on lateral exchange fluxes between stream and surrounding riparian area over the course of the stream, i.e. we cannot conclude if the DOC that is released at the upper catchment is still in the stream at the outlet of the lower catchment. Here, DOC quality

characterization and the investigation of exchange fluxes would be helpful. DOC export is strongly dependent on antecedent moisture conditions, which control the development of soil saturation and in turn hydrologic connectivity to streams. Different topographic positions react differently to precipitation inputs over the course of an event due to the influence of hydrological processes, which define the evolution of connectivity between DOC source zones and the streams, either facilitating or inhibiting DOC transport to the stream. As the frequency and intensity of droughts is likely to increase in the future due to

climate change (Pachauri and Mayer, 2014), our study highlights that the relative contribution of different catchment parts to DOC export from mountainous catchments may change. Especially the importance of riparian zones for DOC export might decrease as hydrological connectivity would be interrupted more often and therefore inhibit DOC mobilization. Longer drought periods could possibly reduce DOC export and slow down the current trend of rising DOC concentrations in freshwater systems.

**Data Availability**

Data supporting the findings of this study are available from the corresponding author upon reasonable request.

**Author contribution**

The study was conceptualized with contributions from all co-authors. KB collected the field data with the support of BB and analyzed the data. KB, LH, BSG, JHF, SP and BB discussed and interpreted the results. KB prepared the manuscript with

contributions from all co-authors.

**Competing interests**

The authors declare that they have no conflict of interest.



**Acknowledgements**

This research was funded by the Rudolf and Helene Glaser-Stiftung in the frame of the project "Influence of natural factors on
concentration, quality and impact of dissolved organic carbon in the Bavarian Forest National Park" (Project No.
T0083\30771\2017\kg). The authors would like to thank the Bavarian Forest National park (BFNP) administration for
providing physiographic and meteorological data, as well as the BFNP staff for their helpful assistance with the installation
and maintenance of field equipment. The authors are also thankful for the personal and technical support obtained through the
project "AquaKlif - Influence of multiple climate-change stressors on stream ecosystems" of the Bavarian Climate Research
Network BayKlif and would also like to thank Ilja van Meerveld and Giulia Zuecco for providing the MATLAB scripts for
the calculation of the hysteresis index.



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
