# Peer review of "Low hydrological connectivity after summer drought inhibits DOC export in a forested headwater catchment"

_Hydrology and Earth System Sciences, 2021_

## Author Comment (AC1)

We appreciate the review by Referee #1 and are grateful for her/his helpful comments, which will certainly improve the manuscript. In the following, we will address the general and the specific comments and explain our intended changes.

**General comments**

This study assesses the controls on the export of dissolved organic carbon (DOC) using high frequency discharge and DOC times series datasets across nested watersheds with contrasting topography. Specifically, the authors focus on event-scale export patterns across four events with generally similar event size, but contrasting antecedent hydrologic conditions.

While the results contribute to our general understanding of DOC export behavior and possible controls, this manuscript can benefit from major revisions that focus on a few areas: (1) clarity – the data in this manuscript is extensive, which while useful, makes it very difficult to follow the Results and Discussion sections. The manuscript would benefit from re-writing certain sections of the manuscript to make the event descriptions and comparisons more clear – see specific comments below for more details. (2) The role of seasonality. While one of the major findings is that events in May and September behaved differently, even though event size was similar. However, the authors do not discuss the role of seasonality in their hierarchy of controlling factors. This seems to miss an important biological control on DOC availability.

There are a range of other specific comments outlined below. Once the authors address these major revisions, I believe the manuscript may be suitable for publication in HESS.

We agree that the data is extensive and therefore some sections might be hard to follow. We will revise the sections mentioned to enhance the clarity of the manuscript according to the comments provided and agree that the manuscript will greatly benefit from this.

Moreover, we agree with both referees that the role of seasonality is an important point. As biological activity is strongly influenced by temperature, DOC production is expected to be higher during the summer months often leading to an increased DOC export. However, this effect seems to be offset at our study site by the pronounced drought period inhibiting hydrological connectivity. We will discuss this in more detail in the revised manuscript.

**Specific comments**

L65- 67 – In addition to event-scale dynamics not being linear or having hysteretic loops, they also often do not mirror annual scale dynamics. Here is a recently published paper that discuss differences in event-scale vs annual scale c-Q relationships that may be relevant for this study:

Fazekas, H. M., Wymore, A. S., & McDowell, W. H. (2020). Dissolved organic carbon and nitrate concentration€• discharge behavior across scales: Land use, excursions, and misclassification. *Water Resources Research*, *56*, e2019WR027028. https://doi.org/10.1029/2019WR027028

Thank you for this recommendation. We will add a sentence about the difficulty of gaining information about annual processes through composite hysteretic loops.

Introduction – the knowledge gap for this study is not well explained. The authors state in L 87-89 the main goal of the study, but don't give necessary motivation leading up to this as to

why this is needed. The paragraphs leading up to this are largely explaining what our community knows about about DOC-Q relationships, but don't address the gaps.

We will further elaborate the knowledge gap for this study. The setting in the National Park allows us to investigate DOC export mechanisms in a region with little anthropogenic influence, which is important against the background of rising DOC concentrations in freshwater systems. There are few studies focusing on the influence of topography on the DOC mobilization mechanisms. Another aspect of increasing importance is the possible impact of climate change and, associated therewith, prolonged drought periods on DOC export.

Description of events – While four events is not that many, it is difficult to keep track of which event is which and how the responses across the watersheds vary. I strongly recommend the authors think about a way to describe these events besides using their dates. For example, could the authors order them by driest (antecedent-wise) to wettest?

We reflected on several options to describe the four events. However, we are of the opinion that the current description using the event dates is useful as we discuss not only the antecedent conditions but also event size, which would be missing if ordering them by driest to wettest. As all events happened at another time of the year, every month is used only once, which prevents confusion. We argue that an order by antecedent conditions would not necessarily make it easier to follow which event is which.

Figure 1 – How were the blue streams in this map determined?

The location of the streams was derived from a DEM with a resolution of 5m. We show all tributaries with a Strahler stream order number > 2.

L 164-165 – To understand the antecedent hydrologic conditions of the four events, it would be helpful if the authors provide antecedent groundwater levels, or the cumulative precipitation from the water year, or some additional information to help the reader understand the context of the event within range of hydrologic conditions that occur in this watershed. Otherwise, there is no clear rationale for why these four events were chosen to represent c-Q dynamics at this site.

We have provided antecedent groundwater levels in Figure 3. In order to clarify the rationale for the selection of the four events, we will refer to Figure 3 earlier in the text in the end of section 2.1. additionally to section 3.1. We will rename section 3.1. to "Hydrological preconditions and discharge behavior" to better summarize the content of the section.

Section 3.3 – This section is extremely hard to follow as written. It is different to understand the differences between all the events at the two different locations. I recommend the authors re-write this section to more clearly introduce the event characteristics.

We are not sure if the referee is really referring to section 3.3. We suppose she/he is referring to section 3.1., where we introduce the event characteristics. We will rewrite this section and better incorporate the groundwater level data as also suggested by Referee #2.

Figure 4 – Have the authors considered calculating runoff ratios? These are good indicators of how much precipitation translates to discharge each event and can help explain c-Q patterns. Further, are the linear regressions necessary? There are so few points, what do the regressions add?

[Figure]

We will add the plot of runoff ratio vs. $AP_{14}$ to Figure 4. We see that Q generation increases with increasing antecedent wetness, which supports our general argument that hydrological connectivity is of importance for Q generation. We will add further explanations to section 3.1. and take up the arguments in the discussion. We are aware that it is difficult to interpret the linear regressions in its original sense due to the low sample size. However, we are of the opinion that they help to recognize a general trend more easily and will therefore not remove them.

Figure 5 – What are the time units on precipitation? Is this precipitation per 15 minutes? Further, it would be helpful for comparison between watersheds, since the watershed sizes are different, to have the discharge area normalized for these analyses.

The precipitation unit in Figure 5 is mm/hr, which we will add to the figure. We will also change discharge to area normalized discharge in mm/h in this figure.

Figure 5 – continued – it is difficult to see the event dynamics in each sub-plot. The authors should consider shortening the x-axis time interval that is displayed to allow readers an opportunity to really see the event specific dynamics.

We will shorten the x-axis to a time interval of 14 days prior to the event, which corresponds to the used $AP_{14}$. This will allow the reader to both see the event specific dynamics in more detail and at the same time be aware of the hydrological preconditions of the event.

Figure 6- Could the authors include an identifier of the antecedent conditions or total P associated with each event? This could go in the upper right corner of each subplot.

Thank you for this helpful suggestion, which we will include in the next version of the figure.

Figure 6 continued – While the caption describes what the color gradient refers to, it would be helpful if the authors include a legend/scale bar. Therefore, the reader would know what color is related to the peak of the event, for example. Otherwise the color gradient only helps identify the start and end of the event.

As suggested by Referee #2 we will add arrows to facilitate the interpretation of the hysteresis loops. However, we will refrain from adding a color gradient as this will add a lot of additional information to the figure, which can be seen without the color gradient as well. The peak of the event, for instance, corresponds to the highest Q value.

L 285 – Can the authors provide more detail about how this 0.32 is calculated? Is this dividing the watershed area between the two watersheds? I do not believe this is described in the Methods section, and for clarity I recommend including this analysis explanation in the Methods section.

When comparing the contribution of different sub-catchments to total Q and DOC export, we assume an equal area contribution of all catchment parts. The calculation of the area ratio is as follows: areaMG/areaHS = 1.1 km$^2$/3.5km$^2$ = 0.31. Therefore, we use the value 0.31 as a benchmark. We will add a sentence to the end of section 2.3. explaining the calculation and adjust the value to 0.31 as we show rounded numbers for the catchment area.

L 305-307 – The authors should back up this statement regarding the relationship between watershed area and event response with literature that has shown this pattern as well. For example, are there studies that have looked at transit time distributions as a function of watershed area? This may help support your argument that water must travel further, and thus takes longer, to reach the watershed outlet.

We will add references supporting our statement.

L 309-311 – The transmissivity feedback concept is relevant in all soils, thus it is unclear why the authors invoke this as a particularly important process in the lower watershed.

We agree that the generally declining saturated hydraulic conductivity in soils could potentially evoke a non-linear increase in lateral subsurface flow in most soils. However, a precondition for this to happen is that the upper soil layers can fully saturate during an event, which usually requires shallow groundwater tables that can quickly rise into the upper soil layers. Such conditions typically exist in flat riparian areas with large TWIs as we only find them in the lower part of our studied catchment. We will add a clarifying sentence to section 4.1.

L 318-320 – Recent work by Michael Rinderer exploring the role of topography on groundwater levels in geographically proximal locations to this study may provide some support for the mechanisms discussed in this section.

We thank you for this valuable suggestion and will include it in section 4.1.

Section 4.1 – Is it possible that there is more groundwater recharge in the lower catchment; that is, as water is transported from a topographic steep landscape to a low gradient landscape, could there be water lost to recharge groundwater at that transition? This could explain why the upper catchment is contributing more flow and DOC relative to the downstream catchment. Alternatively, is it possible that the upper catchment is dominated by shallow stormflow contributions, while the lower catchment is dominated by slower moving deeper groundwater contributions? I believe both of these mechanisms are suggested in the Zimmer and McGlynn (2018) paper cited in this section.

We thank the referee for these valuable suggestions. We do think that both mechanisms could be of importance in the catchment. We also investigated possible groundwater gains or losses along the stream using tracer experiments and radon data. The data indicate that exchange with groundwater is especially important in the lower part of the catchment. However, we decided not to include this data as it is beyond the scope of this study. Nevertheless, we will include the suggestions into section 4.1. and discuss the possibilities of groundwater recharge and contributions in more detail

L 400-404 – There is no mention of the timing of events within this hierarchy of controlling factors. Certainly conditions in biological activity, temperature, etc that vary by season play an important role in DOC export. The authors even discuss this in the previous paragraph. However, it is not mentioned in this concluding paragraph, which seems to therefore miss a critical controlling factor.

As mentioned above we will explain the influence of seasonality in more detail and will include this information in the concluding paragraph as suggested.

**Technical comments**

L 3627 – Is Drake et al 2018 related to the previous sentence? If so, I would recommend moving the citation up a sentence.

As the citation is related to both sentences, we decided to merge the sentences.

L 55-56 – Put "e.g. precipitation" in parentheses.

We will add the parentheses as suggested.

L 196 – should "(1990 and 2010)" be "(1990-2010)"?

Yes, thank you. We will change this.

L 197 – should "compared to 1600 mm" be "compared to long-term average of 1600 mm"?

Yes, we will add this.

L 315 – Missing Figure reference

We will correct this.

Katharina Blaurock

On behalf of all co-authors

---

## Author Comment (AC2)

We appreciate the detailed and very constructive comments by Referee #2, which will greatly improve the manuscript. In the following, we address the general and the specific comments and explain our intended changes.

Blaurock et al. investigated the mobilization of DOC during storm events in two nested, forest catchments in southeast Germany: a 3.5 km$^2$ catchment that includes flat and wide riparian areas at lower elevations, and a smaller and steeper 1.1 km$^2$ catchment upstream. For that, they analyzed a number of metrics and parameters associated with four rainfall events distributed along a ca. two-year period, in which they had high-frequency (15 min) measurements of precipitation, discharge, and DOC concentrations. They conclude that antecedent wetness conditions and topography are major determinants of DOC mobilization.

The topic is definitely interesting and fitted for the audience of Hydrology and Earth System Sciences. The paper is more or less well-written, but at times lack clarity and the reading is not always fluent. I am in general supportive of the interpretations made and of the publication of the paper, but I have many questions, comments, suggestions, and a few concerns that will need to be addressed by the authors before acceptance. Hopefully, these can also help with the presentation issues. Below, I list all my considerations and I look forward to reading the author responses and learn more about this interesting story.

**General comments**

In general, I very much agree with the interpretations made by the authors, but I wonder whether some of them should be toned down given the low sample size (N = 4) and the lack of statistical tests supporting the claims. I appreciate the difficulties of gathering all the appropriate data for a large number of events and the further difficulties to perform meaningful statistical tests with a low sample size, but given that there are statements were parameters are claimed to be higher/lower between the two sites, or being dependent/independent of each other, I wonder whether some statistical analysis can be made to support these claims. What about some simple or multiple linear regressions between parameters or some simple comparison of parameter means between the two sites? I don't imply that any of this should be done, but if not, the authors should justify why no statistical analyses were made and warned the reader that interpretations and based on the hinted evidence.

We aware of the fact that the interpretations are based on a low sample size. We use the figures and regression lines to underline relationships between the investigated parameters. However, we refrain from further statistical analyses as this would not be very reliable nor helpful for further analyses. We will add a sentence about the limitations due to the low sample size in the end of section 2.1.

I agree with a previous reviewer regarding that seasonality is largely disregarded. Two of the studied events happened in spring and the other two in autumn. DOC concentrations in the soil solution and thus in the stream are likely higher in autumn, as shown for other temperate catchments. Do you have an idea if this is the case in your catchment and what role this phenomenon can play in your results? Even if your DOC mobilization is transport-limited and not source-limited, seasonality should still play a role and it has been barely touched (maybe only slightly in LINE 392-393).

We agree with both referees that the role of seasonality is an important point. As biological activity is strongly influenced by temperature, DOC production is expected to be higher during the summer months often leading to an increased DOC export. However, this effect seems to

be offset at our study site by the pronounced drought period inhibiting hydrological connectivity. We will discuss this in more detail in the revised manuscript.

The wordings "antecedent hydrological conditions" and "antecedent wetness conditions" appear mixed in the text and my impression is that they are used interchangeably. I don't think they are analogous terms and in the context of the study I find more appropriate to only use "antecedent wetness", as you are using antecedent precipitation as a proxy for wetness and not for hydrological conditions (precisely because, as you argue in the paper, event size is not a good predictor of discharge).

We agree that the use of only "antecedent wetness conditions" will prevent misunderstandings and will change the wording accordingly.

I think all discharge data presented in the paper should be normalized to catchment area, i.e. presented in units of mm. This would allow comparing discharge more easily between the two sites and with other sites.

We will change the discharge data to normalized data according to the suggestion made by both referees.

I find the parameter "DOC load (kg)" largely irrelevant and would remove it together with all the related results and discussions. I would actually change it to "Area specific DOC load (kg $m^{-2}$)", which is a lot more meaningful.

We decided to use the unit [kg/hr], which is a unit typically used for load, instead of [kg]. This will help us to put the number in the context of the specific event and to prevent a bias linked to the duration of the events. We will adjust the related results and discussions. The reference to area is already included in the precipitation specific DOC load [kg $km^{-2}$ $mm^{-1}$].

While the use of sensors has allowed obtaining high-frequency data, the measurements obtained with sensor loggers are not "continuous" but respond to a fixed-interval. Please, correct the few instances where "continuous measurements" were mentioned and simply specify their frequency or that they were highly-frequent.

We will change this according to the reviewer's suggestion.

I would define catchment Markungsgraben as "MG" and catchment Hinterer Schachtencbach as "HS" sooner in the text, and then present them, when possible, always in the same order.

We will add the explanation of the abbreviations to section 2.1 and check the order throughout the manuscript.

Throughout the manuscript, both the term "watershed" and the term "catchment" are used. I would use only one of the two, preferably "catchment".

We will change the wording as suggested.

**Specific comments**

Title

The word "Connectivity" is too vague in the context. I would rather say "hydrological connectivity". I am also a bit sceptical about the word "missing". Maybe a better word is simply "low"? Finally, I would emphasize that the mobilization was studied during rainfall events. What about then: "Low hydrological connectivity during summer controls DOC mobilization and export during rainfall events in a small, forested catchment"? Or something similar.

We will change the title as follows: „Low hydrological connectivity following summer drought inhibits DOC export in a forested headwater catchment"

Abstract

LINE 10. DOC needs to be defined.

We will add the definition to the Abstract.

LINE 11. "hypothesized" instead of "hypothesize".

We will change this.

LINE 11. In which contexts is topography a key driver of DOC export? Please, specify (e.g. in headwater catchments).

We will add "in headwater catchments" as suggested.

LINE 12. I would rather use "hydrological" instead of "hydrologic", or at least only one of the two terms throughout the paper. Now they appear to be mixed.

We will change the wording as suggested and stick to "hydrological" throughout the manuscript.

LINE 12. Maybe you better mean "To test this hypothesis"?

We will change this.

LINE 14-16. I don't think this is the best way to describe where the measurements were done. Discharge and DOC were measured in two stream locations, not in a steep hillslope or a flat riparian zone as the sentence as written now implies. Please, rephrase this part to make clear that the measurements were done in the stream, maybe specifying that at one of the locations the stream drains a steep area, whereas at the other location it drains a bigger area that includes a flat and wide riparian zone at lower elevations.

We agree that the description might be misleading and will rewrite this part to clarify the location of the measurements.

LINE 17. By "During events" you mean during the four studied events? I think so and if so, please specify it.

We will add "during the events", which refers to the events mentioned in the sentence before.

LINE 21. This number (522 kg) is largely uninformative without a reference, which in this case I think it should be a normalization to catchment area (see my general comment related to this issue).

As explained above, we prefer to use the reference to a time (the length of the event).

LINE 23. Rather than "lack of hydrological connectivity" I would say "low hydrological connectivity", as the stream is still receiving water from the surrounding catchment area. As I understood, there is no evidence suggesting that the stream is completely disconnected from the catchment under dry conditions, losing water towards the riparian zone (right?). But if there is a complete hydrological disconnection, it should be explained.

We do not have evidence that there is a complete disconnection and will therefore tone down the statement by using "low hydrological connectivity" as it will then be used in the title.

LINE 27. I wonder whether there is a better word than "parts" in this context. Maybe "locations" or "compartments"?

We will use the term "sub-catchment" as we already use it in other parts of the manuscript.

LINE 28. Similar to the comment on LINE 23, hydrological connectivity will still occur in the future (unless the stream completely disconnects from the catchment, which I assume it is not the case, not even in summer), only that its degree will be lower depending on the conditions. Thus, I would say something like "will be reduced" or something similar.

We will change this statement to "when hydrological connectivity will be reduced more often".

1 Introduction

LINE 36. Please, move the citation to Drake et al. (2018) to the end of this sentence.

As the citation is related to both sentences, we decided to merge the sentences to one.

LINE 42. The conclusions drawn by Freeman et al. (2001) were admittedly questionable and I would suggest not to cite this paper.

We will remove the citation.

LINE 43. "influences terrestrial carbon pools". How? By depleting them? Please, specify.

We will change the sentence to "Rising DOC concentrations indicate an increased leaching from soils and peatlands and have the potential to deplete the terrestrial carbon pools, which are of global importance for carbon storage".

LINE 50. Please, note that a reduction in ionic strength is not an independent process but rather a consequence of a decline in atmospheric acid deposition. Thus, it does not fit in this list.

We will move the reference of Hruška et al (2009) to the other references referring to a decline in atmospheric deposition.

LINE 48-54. In this context, I would suggest having a look at Clark et al. (2010), who nicely summarized the potential factors behind rising DOC concentrations (which have not really changed since that paper was published) and who importantly highlighted that these factors operate on varying temporal and spatial scales. This might be more relevant to your study, although this topic is in general tangential to what it is investigated.

We will add a sentence about the difficulty explained by Clark et al. (2010) concerning the differences in spatial and temporal scales of the studies.

LINE 62. I would write "which can then be mobilized as DOC", rather than " including DOC, which is easily mobilized".

We will change this as suggested.

LINE 72-74. This part of the sentence seems incoherent with respect to the first part of it. Please, rephrase.

We will change the sentence as follows: „Anti-clockwise hysteretic loops usually indicate a delayed arrival of DOC at the stream, which can be caused by the source areas being located further away from the stream (Hood et al., 2006; Vaughan et al., 2017), the sources being connected via flow paths with slow transport velocities (Musolff et al., 2017) or by changes in the dominant flow paths and associated changes in hydrological connectivity (Brown et al., 1999; Hagedorn et al., 2000; Schwarze and Beudert, 2009; Strohmeier et al., 2013; Cerro et al., 2014, Ågren et al., 2008; Pacific et al., 2010)."

LINE 79. Please, remove "itself".

We will remove "itself".

LINE 79. Does "appears" refer to the beginning of the sentence, i.e. to "Hydrological connectivity". If so, please add commas in between "and therefore […] McDonnel , 2010)".

We will add commas after connectivity and response to mark "and therefore runoff and solute response" as an additional information.

LINE 82. I would write "DOC" instead of "C".

We will change this.

LINE 89-91. This sentence should be written in past tense, as the hypotheses should define your expectations prior conducting the experiments.

We will change the tenses used here.

LINE 93. I am still not satisfied with the wording "parts of the catchment". Maybe write "between sub-catchments dominated by either of these two topographical configurations", or something similar.

As explained above, we will use the term "sub-catchment".

2 Material and Methods

LINE 105. Maybe it is better to mention here that the Kaltenbrunner Seige sub-catchment was not explicitly studied in this paper. It is also probably better not to mention this catchment again to avoid adding unnecessary unfamiliar names for the reader to keep track.

We will state that the sub-catchment Kaltenbrunner Seige was not studied already in section 2.1. and try to reduce the mention of this sub-catchment to the minimum.

LINE 123-130. This information can be presented in a more clear and simplified manner. I would just mention that you have one sampling location close to the outlet of the Markungsgraben catchment at an elevation of 888 m a.s.l., and that this location would be referred thereafter as MG. Briefly say that this catchment is steep and refer to Table 1. Then mention that the second sampling location is close to the outlet of the Hinterer Schachtencbach catchment at an elevation of 771 m a.s.l., and that this location would be referred thereafter as HS. Briefly say that this catchment drains flatter areas with wide riparian zones at lower elevations. I would avoid presenting any other information.

We agree that some of the information in the text is unnecessary and will therefore follow the suggestion made by the referee.

LINE 132. At what resolution? Please, specify.

All data used for the long-term mean values were measured at a daily resolution. We will add this information.

LINE 134-136. What was this done for? What is the aim of this in the context of the study?

We will change the sentence to: „In order to assess the general meteorological conditions during the sampling period 2018, 2019 and 2020, long-term mean monthly values for the two stations for the period from 1990 to 2010 were calculated." We further explain the characteristics of the sampling periods in section 3.1.

LINE 138-141. The three locations where groundwater level data was monitored should be included in the map of Figure 1. As it is described now, it is difficult to know where they were located with respect to the stream measurement locations. For example, what does "uphill" mean? How far from streams where these three groundwater monitoring stations located, and in which type of soil? In any case, the integration of these data into the story of the paper should be improved. As they are presented now, they do not appear very relevant.

We will include the locations in the map of Figure 1 and add some additional information about the depth and soil characteristics. We use the groundwater level data to characterize the long-term hydrological conditions of the catchment throughout the sampling periods. As we describe in section 3.1., we can distinguish between two events following dry periods with declining groundwater tables and two events following higher groundwater tables after snowmelt.

LINE 149. What was the resolution of the discharge measurements from the MG site? Given that comparing discharge and exports between the two locations was a major aspect of the study, consideration should be given to the uncertainties associated with the discharge measurements, especially when you have two sources of data with different resolutions. How confident are you that the two discharge time series from the two stream locations can be directly compared?

For both locations, the resolution of the discharge measurements was 15 minutes. We are therefore sure that the discharge time series can be directly compared. We agree that the description was confusing and will clarify this sentence and add the information about the resolution for MG.

LINE 155. So, the grab sample values were added to the software in order to update the internal calibration into a so-called "local calibration", right? This is critical, as I wouldn't trust the default calibration.

We did not add the values to the software by using a "local calibration" but adjusted the default calibration afterwards by using the values measured in the laboratory as we are aware of the fact that the default calibration is not completely reliable. We will therefore add: "In order to refine the internal calibration, the DOC concentrations measured…".

LINE 160. Any reason why the DOC calibration for MG was not as good as the calibration for HS?

We are not able to explain why the DOC calibration for MG was not as good as for HS. Due to a technical failure, we had to replace the spectrolyser at MG in July. Therefore, the calibration for the event in September 2020 is different with a $R^2$ of 0.97. This suggests that the mediocre calibration is linked to the specific device. We will add the information about the different calibrations at MG to section 2.2.4

LINE 163-165. It feels like this sentence would fit better in the next section. In any case, this part has to be better presented and justified, as it is the basis of all subsequent analyses. Why these four events? What criteria were followed to select them? How do they compare with other events during the study period? Why no other events were included?

As the sampling period was very dry, not many events were available for analysis. Only some small events and very few large events could be observed. Small events led to small discharge and/or DOC responses or no responses at all, which would make the analysis of hysteresis patterns difficult, for instance. We decided to focus on the largest events in order to be able to analyze DOC responses in the stream in detail. We will add a sentence about the reasons for selecting the four events presented in the manuscript in order clarify this.

LINE 167. For this first sentence to be compelling, first you would need to describe how baseflow was classified. Thus, I would move the sentence to a later point, after you have described how you define events.

We will restructure this paragraph as suggested.

LINE 176. The 15-min resolution values, right? Please, specify it.

Yes, we will add the information about the resolution.

3 Results

LINE 196. Please, write "1990-2010" instead of "1990 and 2010".

We will change this.

LINE 197. Do you mean "compared to the long-term average of 1600 mm"?

Yes, we will change this accordingly.

LINE 198-200. I would start the paragraph with this sentence instead.

We will change this as suggested.

LINE 195-200. I wonder how relevant this information and Figure 2 are for the paper. If it is just to put you study period into a long-term context (weather-wise), I would consider removing it, at least the figure. Otherwise, please integrate this part better into the story.

We think that the information is relevant to put the study period into a long-term context, as the study period was particularly dry. However, we agree that the figure is not necessary as the important information is given in the text. We will therefore remove the figure.

LINE 204-206. This part related to the groundwater tables (including Figure 3) should also be better integrated into the story. In any case, I am a bit puzzled by what I see in Figure 3. To me it appears that, in general, groundwater tables do not really react to any of the studied events. Is there any reason for this? Where are the groundwater monitoring station located? It seems like soils are very deep there.

The data is representing the deep groundwater. As it can be seen in Figure 3, the groundwater level varies between 2 and 16 meters below ground. In our opinion, it is therefore not surprising that we observe seasonal variations only instead of a response to events. As explained above, we will add the location of the groundwater wells to Figure 1 and add some additional information.

LINE 218-221. This part feels like it belongs to the discussion.

We think that this information is useful at this part of the manuscript to explain the reasons for studying the selected events and to highlight the differences between them.

LINE 239-240. I don't know what it is meant here. If you want to refer to the baseflow periods immediately prior the four events, please describe it explicitly.

As this information is not really relevant for the study, we decided to remove this sentence.

LINE 242. "without a clear relation". Did you plot this?

We plotted the relation but decided not to include it in the manuscript in order to focus on other points. However, we will add a reference to Table 2, where all values are presented.

LINE 262. "where concentrations decreased soon after reaching the DOC peak". I assume this refers to MG, and not to HS nor to what it is written in parenthesis, but the way the sentence is written makes it confusing. Please, rephrase.

We will change the sentence to: "This resulted in wider hysteretic loops at HS than at MG (larger absolute values of $h$) as the concentrations at MG decreased soon after reaching the DOC peak. "

LINE 283. It is unlikely, but a good theoretical approximation. I would leave this for the discussion, and here just say that you assume equal area contribution.

We prefer to leave the sentence as it is. However, as suggested by Referee #1, we will add some more detailed information about how we derived this value.

LINE 290. I realize that the different panels of Figure 4 are not presented in the natural order (a to f) within the results. Could you please either reorganize/relabel the figures or the text to present them in order?

We agree and will reorganize this section.

4 Discussion

LINE 300. But is this driven by P or by $AP_{14}$?

As explained in the following sentence, precipitation alone is not the main driver of Q generation but the antecedent hydrological conditions are of importance.

LINE 306. Please, rewrite this sentence as it is unclear.

We will change the sentence to "To some extent, this observation can be attributed to the larger catchment area contributing to water fluxes at HS, resulting in longer flow pathways and a delayed Q response." and add two references to back up this statement.

LINE 312. In which way is hydrological connectivity the driver here? Please, make it explicit at this point, or mention that you will explain it in the following paragraph.

We will change this paragraph as follows: "We suggest that hydrological connectivity between the wide riparian zone and the stream is the major driver for delivering water to the stream. The hydrological connectivity is dependent both on topography as on the antecedent conditions as we will explain in the following."

LINE 315. The figure number seems to be missing.

We will add the missing figure number.

LINE 321. Please, change "starts sooner" by "is faster".

We will change this as suggested.

LINE 323-324. This needs to be better explained. What kind of "lowlands" and "headwaters" did Zimmer and McGlynn studied and where? Briefly specify it and make the connection to your study.

We think that the sentence does not fit well here in general. We will remove it and discuss the results by Zimmer and McGlynn in more detail in section 4.2.

LINE 332-333. This might be the "expected" range for forested catchments in temperate regions, but it is not the normal range for e.g. boreal, Mediterranean, or tropical sites, so please

specify your ecoregion. Also, I would change "expected" by other wording such as "comparable with" or "similar to".

We will change the sentence as follows: "Concentrations were similar to values found in other temperate forested catchments in low mountain ranges."

LINE 333-334. "Larger events generally lead to higher DOC concentrations in streams". Are you referring to your study or to other studies? If the latter, please add a reference. If the former, please remind the reader how you showed this.

We will add a reference to back up this statement.

LINE 336-339. This is an important conclusion, but it is not universal. To make it more broadly relevant, please argue in what contexts might be applicable.

We agree that we can discuss the broader context in more detail. However, we prefer to do this in the Conclusions section, where we discuss possible implications of climate change for the relative contribution of different sub-catchments. There, we will add a sentence regarding the relevance of transport limitation in the context of climate change.

LINE 354-356. Maybe remind the reader that you can make this claim because in this catchment DOC appears to be transport-limited rather than source-limited.

We will add the suggestion as follows: "As DOC appears to be transport-limited rather than source-limited, the persistently high concentrations, in combination with a high discharge generation due to the existing hydrological connectivity, could then cause the pronounced DOC export during events following wet antecedent conditions."

LINE 357-360. I don't know if I agree with the way the transmissivity feedback mechanism is invoked here. The mechanism explains the fast, but deaccelerated increase in groundwater tables due to the saturation of highly conductive shallow soil layers. Thus, at the beginning of an event the increase of groundwater tables would be fast, and then would slow down due to the activation of the highly conductive layers that have a higher lateral water transfer rate. How does the mechanism really connect to your findings? How deep are your soils and how does the groundwater table behave during events? This is where the groundwater table data can be useful.

We do not think that the groundwater table data shown here can be used to investigate the transmissivity feedback, as this is a process occurring in the upper soil layer. The groundwater data shown here, however, refers to the deep groundwater level representing slow changes of the groundwater table. However, we also installed piezometers in the shallow groundwater later during the sampling campaign. There we do see the relationship between the groundwater table and discharge as explained by the Referee. We do not include this data as it cannot be linked to the selected events of this study due to different sampling periods and will be part of a different manuscript. Nevertheless, we think that the transmissivity feedback can be of importance in the riparian zone of the lower part of the catchment because there we do see shallow groundwater tables, which can quickly rise into the upper soil layers.

LINE 363. Why later during the event?

During dry periods, those pools are empty and start filling only with the beginning of the precipitation event. They connect to the stream once a certain water level is reached and therefore contribute to discharge later during the event. We will therefore change the sentence as follows: "The possibility that these pools contribute to DOC export when filled with water later during the event is…".

LINE 366-369. These explanations are critical in the study, but I am not sure I fully understand them in light of the results. Wouldn't this process imply clockwise hysteresis loops instead of anti-clockwise loops. Why is the activation of sources so slow in your catchment? As I understand it, you are implying that there is a relationship between antecedent wetness and type of hysteresis, but from the data presented in Table 2 and Figure 4b, it doesn't look like there is a relationship between wetness and "h index" in the HS catchment. This point needs to be carefully addressed.

Although we compare only four data points, we do think that a relation is visible in Figure 4b. We observe smaller hysteresis loops (h closer to zero) during wet conditions than during dry conditions. The event in October, following the dry summer, shows the broadest loop, the event in June the smallest. It is not unusual to only find anti-clockwise hystereses when comparing our results to other studies. Anti-clockwise hysteresis loops are caused by first having to activate the most potent DOC sources in the shallow soils by bringing the water table up to hydrologically connect them with the stream during the rising limb of the event hydrograph and during the falling limb the DOC rich upper soil layers are still draining while the discharge recedes. The activation of sources is generally slow in our catchment but seems to be accelerating if a certain hydrological connectivity is present. We will add an explaining sentence at the end of section 3.2.

LINE 370-374. The contrast with other studies in this sense might be also explained by the fact that DOC is transport-limited rather than source-limited, as you argue.

We do not think that the transport-limitation in our catchment is in contrast to other studies as many catchments are transport limited. To further stress the importance of transport limitation, we will change the sentence as follows: "If the soils are wet prior to an event, connected flow paths can quickly be established and DOC transport to the stream occurs faster than during dry conditions, which highlights that DOC export is transport limited in this catchment."

LINE 410-411. Precisely, as I commented in LINE 366-369, I don't see this pattern in Figure 4b. If I understood it correctly, there might be a weak relationship between catchment wetness and h index for the MG site, but not for the HS site. Is there any type of error in the figure? I might be misunderstanding something, but if the figure and values shown in Table 2 are correct, this part needs to be corrected and some of the discussions you present need to be reconciled with this observation, which is the opposite of what you arguing now.

As described above, we do think that there is a (admittedly not very strong) relation between catchment wetness and h index. However, we agree that the paragraph needs to be restructured in order to clarify our arguments. We will rewrite it accordingly.

LINE 417-419. Where and in what type of catchment did Correa et al. (2019) made this observation.

We will add this information: "Correa et al. (2019) made a similar observation in a tropical alpine headwater catchment with anti-clockwise hysteresis..."

LINE 425. I would end the sentence with "[…] a higher general wetness that favours the build up of DOC in the soil" and would add a reference.

We will change the sentence to: „According to these studies, a flat area would therefore tend to export more DOC than a steep area due to a higher general wetness that favors the build-up of DOC in the soil and hydrological connectivity."

LINE 429-431. I think this sentence is largely irrelevant and I would remove it.

We will remove the sentence as suggested.

LINE 442. "is proportionally higher". Already taking into account the differences in precipitation between the two locations? Please, specify.

We will change the sentence as follows: "The proportional amount of Q arriving from the upper catchment is higher than during the other events; however, the DOC proportion is even higher."

LINE 439-454. The addition of a column in Table 1 with the same information for the entire HS catchment would help interpreting and supporting these explanations.

We agree that the information would be useful and will add the column in Table 1 but remove the information about the entire Große Ohe catchment as suggested further down.

LINE 463-464. Which in a way is a specific case of the previous explanation, rather than another different reason. Please, reformulate.

We will rephrase the sentence as follows: "A consequence of the reduced connectivity of the lower catchment could be that the riparian pools mentioned above are not connected and thus an important DOC source is not active."

5 Conclusions

LINE 477-479. Do you have reasons to suspect that at this stretch of the stream is a net looser of water at any time of the year? Another reason why DOC can be lost is in-stream mineralization. Can this play a role?

We also investigated possible groundwater gains or losses along the stream using tracer experiments and radon data. The data indicate that exchange with groundwater is especially important in the lower part of the catchment. However, we decided not to include this data as it is beyond the scope of this study. Nevertheless, we will discuss the possibilities of groundwater recharge and contributions in more detail in section 4.1. as suggested by Referee 1. We do not think that in-stream mineralization is if importance as the processes investigated happen at the event-scale and are therefore rather fast in contrast to mineralization processes.

Tables and Figures

Table 1. I would remove the column with the information about the entire Grosse Ohe catchment, as it is more distracting than anything else. As I understand, the information presented for the Hinterer Schachtencbach catchment only reflects the local sub-catchment, but I would also like to see the analogous, integrated information for the entire catchment (i.e. for the whole 3.5 km$^2$ that include the other two subcatchments).

As explained above, we will add the column in Table 1 but remove the information about the entire Große Ohe catchment as suggested further down.

Figure 5. Please, add HS and MG on top of the left and right panels, as in Figure 6. Please change to "mm/15 min" the units of the precipitation (if that's the case).

We will add HS and MG as suggested and add the units of precipitation (mm/hr).

Figure 6. Besides the colour code, a small arrow (or a couple of arrows in the lower panels) indicating the direction of the hysteresis loops in each panel would help visualizing and interpreting the results.

We will add arrows as suggested.

Suggested references

Clark, J. M., Bottrell, S. H., Evans, C. D., Monteith, D. T., Bartlett, R., Rose, R., Newton, R. J., and Chapman, P. J.: The importance of the relationship between scale and process in understanding long-term DOC dynamics, Science of the Total Environment, 408, 2768-2775, 10.1016/j.scitotenv.2010.02.046, 2010.

Katharina Blaurock

On behalf of all co-authors

---

## Author Response (AR1)

**Answer to Referee #1**

We appreciate the review by Referee #1 and are grateful for her/his helpful comments, which helped to improve the manuscript. In the following, we will address the general and the specific comments and explain which changes were made. We refer to the line numbers of the marked up manuscript.

**General comments**

This study assesses the controls on the export of dissolved organic carbon (DOC) using high frequency discharge and DOC times series datasets across nested watersheds with contrasting topography. Specifically, the authors focus on event-scale export patterns across four events with generally similar event size, but contrasting antecedent hydrologic conditions.

While the results contribute to our general understanding of DOC export behavior and possible controls, this manuscript can benefit from major revisions that focus on a few areas: (1) clarity – the data in this manuscript is extensive, which while useful, makes it very difficult to follow the Results and Discussion sections. The manuscript would benefit from re-writing certain sections of the manuscript to make the event descriptions and comparisons more clear – see specific comments below for more details. (2) The role of seasonality. While one of the major findings is that events in May and September behaved differently, even though event size was similar. However, the authors do not discuss the role of seasonality in their hierarchy of controlling factors. This seems to miss an important biological control on DOC availability.

There are a range of other specific comments outlined below. Once the authors address these major revisions, I believe the manuscript may be suitable for publication in HESS.

We agree that the data is extensive and therefore some sections might be hard to follow. We revised the sections mentioned to enhance the clarity of the manuscript according to the comments provided and agree that the manuscript will greatly benefit from this.

We agree with both referees that the role of seasonality is an important point, which has been object of several studies. As biological activity is strongly influenced by temperature, DOC production is expected to be higher during the summer months often leading to an increased DOC export. We do mention this expectation, when discussing the low DOC export during the fall events (L436-443) and again when discussing the relatively higher DOC export of the large September event (L460). As suggested, we added the aspect of seasonality to the hierarchy of controlling factors at the end of section 4.2 (L472).

**Specific comments**

L65- 67 – In addition to event-scale dynamics not being linear or having hysteretic loops, they also often do not mirror annual scale dynamics. Here is a recently published paper that discuss differences in event-scale vs annual scale c-Q relationships that may be relevant for this study:

Fazekas, H. M., Wymore, A. S., & McDowell, W. H. (2020). Dissolved organic carbon and nitrate concentration–€• discharge behavior across scales: Land use, excursions, and misclassification. *Water Resources Research*, *56*, e2019WR027028. https://doi.org/10.1029/2019WR027028

Thank you for this recommendation. We added a sentence about the difficulty of gaining information about annual processes through composite hysteretic loops (L74).

Introduction – the knowledge gap for this study is not well explained. The authors state in L 87-89 the main goal of the study, but don't give necessary motivation leading up to this as to why this is needed. The paragraphs leading up to this are largely explaining what our community knows about about DOC-Q relationships, but don't address the gaps.

We added more information about the motivation for this study (L99-103).

Description of events – While four events is not that many, it is difficult to keep track of which event is which and how the responses across the watersheds vary. I strongly recommend the authors think about a way to describe these events besides using their dates. For example, could the authors order them by driest (antecedent-wise) to wettest?

We reflected on several options to describe the four events. However, we are of the opinion that the current description using the event dates is useful as we discuss not only the antecedent conditions but also event size, which would be missing if ordering them by driest to wettest. As all events happened at another time of the year, every month is used only once, which prevents confusion. We argue that an order by antecedent conditions would not necessarily make it easier to follow which event is which.

Figure 1 – How were the blue streams in this map determined?

The location of the streams was derived from a DEM with a resolution of 5m. We added this information to the caption of Figure 1.

L 164-165 – To understand the antecedent hydrologic conditions of the four events, it would be helpful if the authors provide antecedent groundwater levels, or the cumulative precipitation from the water year, or some additional information to help the reader understand the context of the event within range of hydrologic conditions that occur in this watershed. Otherwise, there is no clear rationale for why these four events were chosen to represent c-Q dynamics at this site.

We have provided antecedent groundwater levels in Figure 2 (former Figure 3). In order to clarify the rationale for the selection of the four events, we now refer to Figure 2 earlier in the text at the end of section 2.2.4. (L197) additionally to section 3.1. We renamed section 3.1. to "Hydrological preconditions and discharge response" to better summarize the content of the section.

Section 3.3 – This section is extremely hard to follow as written. It is different to understand the differences between all the events at the two different locations. I recommend the authors re-write this section to more clearly introduce the event characteristics.

We are not sure if the referee is really referring to section 3.3. We suppose she/he is referring to section 3.1., where we introduce the event characteristics. We rewrote parts of this section to present the event characteristics in a more structured way.

Figure 4 – Have the authors considered calculating runoff ratios? These are good indicators of how much precipitation translates to discharge each event and can help explain c-Q patterns.

Further, are the linear regressions necessary? There are so few points, what do the regressions add?

We now calculated runoff ratios and added the values to Table 2 and do refer to the values in the Results (L286-271) and Discussion (L382). We are aware that it is difficult to interpret the linear regressions in its original sense due to the low sample size. However, we are of the opinion that they help to recognize a general trend more easily and will therefore not remove them.

Figure 5 – What are the time units on precipitation? Is this precipitation per 15 minutes? Further, it would be helpful for comparison between watersheds, since the watershed sizes are different, to have the discharge area normalized for these analyses.

The precipitation unit in Figure 4 (former Figure 5) is mm/h, which we added to the Figure. We also changed discharge to area normalized discharge in mm/h in this Figure.

Figure 5 – continued – it is difficult to see the event dynamics in each sub-plot. The authors should consider shortening the x-axis time interval that is displayed to allow readers an opportunity to really see the event specific dynamics.

We shortened the x-axis to a time interval of 14 days prior to the event, which corresponds to the used $AP_{14}$. This will allow the reader to both see the event specific dynamics in more detail and at the same time be aware of the hydrological preconditions of the event.

Figure 6- Could the authors include an identifier of the antecedent conditions or total P associated with each event? This could go in the upper right corner of each subplot.

Thank you for this helpful suggestion, which we included this information.

Figure 6 continued – While the caption describes what the color gradient refers to, it would be helpful if the authors include a legend/scale bar. Therefore, the reader would know what color is related to the peak of the event, for example. Otherwise the color gradient only helps identify the start and end of the event.

As suggested by Referee #2 we added arrows to facilitate the interpretation of the hysteresis loops. However, we will refrain from adding a color gradient as this will add a lot of additional information to the figure, which can be seen without the color gradient as well. The peak of the event, for instance, corresponds to the highest Q value.

L 285 – Can the authors provide more detail about how this 0.32 is calculated? Is this dividing the watershed area between the two watersheds? I do not believe this is described in the Methods section, and for clarity I recommend including this analysis explanation in the Methods section.

When comparing the contribution of different sub-catchments to total Q and DOC export, we assume an equal area contribution of all sub-catchments. The calculation of the area ratio is as follows: areaMG/areaHS = 1.1 $km^2$/3.5$km^2$ = 0.31. Therefore, we use the value 0.31 as a benchmark. We added a sentence to section 2.3. explaining the calculation (L217-219) and adjust the value to 0.31 as we show rounded numbers for the catchment area.

L 305-307 – The authors should back up this statement regarding the relationship between watershed area and event response with literature that has shown this pattern as well. For example, are there studies that have looked at transit time distributions as a function of watershed area? This may help support your argument that water must travel further, and thus takes longer, to reach the watershed outlet.

We added two references supporting our statement (L358).

L 309-311 – The transmissivity feedback concept is relevant in all soils, thus it is unclear why the authors invoke this as a particularly important process in the lower watershed.

We agree that the generally declining saturated hydraulic conductivity in soils could potentially evoke a non-linear increase in lateral subsurface flow in most soils. However, a precondition for this to happen is that the upper soil layers can fully saturate during an event, which usually requires shallow groundwater tables that can quickly rise into the upper soil layers. Such conditions typically exist in flat riparian areas with large TWIs as we only find them in the lower part of our studied catchment. We could observe this as well using a time-lapse camera and added this information to section 4.1. (L363-366).

L 318-320 – Recent work by Michael Rinderer exploring the role of topography on groundwater levels in geographically proximal locations to this study may provide some support for the mechanisms discussed in this section.

We thank you for this valuable suggestion, which we included in section 4.1. (L376-379).

Section 4.1 – Is it possible that there is more groundwater recharge in the lower catchment; that is, as water is transported from a topographic steep landscape to a low gradient landscape, could there be water lost to recharge groundwater at that transition? This could explain why the upper catchment is contributing more flow and DOC relative to the downstream catchment. Alternatively, is it possible that the upper catchment is dominated by shallow stormflow contributions, while the lower catchment is dominated by slower moving deeper groundwater contributions? I believe both of these mechanisms are suggested in the Zimmer and McGlynn (2018) paper cited in this section.

We thank the referee for these valuable suggestions. We do think that both mechanisms could be of importance in the catchment. We also investigated possible groundwater gains or losses along the stream using tracer experiments and radon data. The data indicate that exchange with groundwater is especially important in the lower part of the catchment. However, we decided not to include this data as it is beyond the scope of this study. Nevertheless, we included the suggestions into section 4.1. and discuss the possibilities of groundwater recharge and contributions in more detail (L370-374).

L 400-404 – There is no mention of the timing of events within this hierarchy of controlling factors. Certainly conditions in biological activity, temperature, etc that vary by season play an important role in DOC export. The authors even discuss this in the previous paragraph. However, it is not mentioned in this concluding paragraph, which seems to therefore miss a critical controlling factor.

As mentioned above, we added the aspect of seasonality to the hierarchy of controlling factors at the end of section 4.2.

**Technical comments**

L 3627 – Is Drake et al 2018 related to the previous sentence? If so, I would recommend moving the citation up a sentence.

As the citation is related to both sentences, we decided to merge the sentences.

L 55-56 – Put "e.g. precipitation" in parentheses.

We added the parentheses as suggested.

L 196 – should "(1990 and 2010)" be "(1990-2010)"?

Yes, thank you. We changed this.

L 197 – should "compared to 1600 mm" be "compared to long-term average of 1600 mm"?

Yes, we added this.

L 315 – Missing Figure reference

We corrected this.

Katharina Blaurock

On behalf of all co-authors

**Answer to Referee #2**

We appreciate the detailed and very constructive comments by Referee #2, which greatly improved the manuscript. In the following, we address the general and the specific comments and which changes were made. We refer to the line numbers of the marked up manuscript.

Blaurock et al. investigated the mobilization of DOC during storm events in two nested, forest catchments in southeast Germany: a 3.5 km$^2$ catchment that includes flat and wide riparian areas at lower elevations, and a smaller and steeper 1.1 km$^2$ catchment upstream. For that, they analyzed a number of metrics and parameters associated with four rainfall events distributed along a ca. two-year period, in which they had high-frequency (15 min) measurements of precipitation, discharge, and DOC concentrations. They conclude that antecedent wetness conditions and topography are major determinants of DOC mobilization.

The topic is definitely interesting and fitted for the audience of Hydrology and Earth System Sciences. The paper is more or less well-written, but at times lack clarity and the reading is not always fluent. I am in general supportive of the interpretations made and of the publication of the paper, but I have many questions, comments, suggestions, and a few concerns that will need to be addressed by the authors before acceptance. Hopefully, these can also help with the

presentation issues. Below, I list all my considerations and I look forward to reading the author responses and learn more about this interesting story.

**General comments**

In general, I very much agree with the interpretations made by the authors, but I wonder whether some of them should be toned down given the low sample size (N = 4) and the lack of statistical tests supporting the claims. I appreciate the difficulties of gathering all the appropriate data for a large number of events and the further difficulties to perform meaningful statistical tests with a low sample size, but given that there are statements were parameters are claimed to be higher/lower between the two sites, or being dependent/independent of each other, I wonder whether some statistical analysis can be made to support these claims. What about some simple or multiple linear regressions between parameters or some simple comparison of parameter means between the two sites? I don't imply that any of this should be done, but if not, the authors should justify why no statistical analyses were made and warned the reader that interpretations and based on the hinted evidence.

We are aware of the fact that the interpretations are based on a low sample size. We use the figures and regression lines to underline relationships between the investigated parameters. However, we refrain from further statistical analyses as this would not be very reliable nor helpful for further analyses. We added a sentence about the limitations due to the low sample size at the end of section 2.3. (L230).

I agree with a previous reviewer regarding that seasonality is largely disregarded. Two of the studied events happened in spring and the other two in autumn. DOC concentrations in the soil solution and thus in the stream are likely higher in autumn, as shown for other temperate catchments. Do you have an idea if this is the case in your catchment and what role this phenomenon can play in your results? Even if your DOC mobilization is transport-limited and not source-limited, seasonality should still play a role and it has been barely touched (maybe only slightly in LINE 392-393).

We agree with both referees that the role of seasonality is an important point, which has been object of several studies. As biological activity is strongly influenced by temperature, DOC production is expected to be higher during the summer months often leading to an increased DOC export. We do mention this expectation, when discussing the low DOC export during the fall events (L436-443) and again when discussing the relatively higher DOC export of the large September event (L460). As suggested, we added the aspect of seasonality to the hierarchy of controlling factors at the end of section 4.2 (L472).

The wordings "antecedent hydrological conditions" and "antecedent wetness conditions" appear mixed in the text and my impression is that they are used interchangeably. I don't think they are analogous terms and in the context of the study I find more appropriate to only use "antecedent wetness", as you are using antecedent precipitation as a proxy for wetness and not for hydrological conditions (precisely because, as you argue in the paper, event size is not a good predictor of discharge).

We agree that the use of only "antecedent wetness conditions" will prevent misunderstandings and changed the wording accordingly.

I think all discharge data presented in the paper should be normalized to catchment area, i.e. presented in units of mm. This would allow comparing discharge more easily between the two sites and with other sites.

We changed the discharge data to normalized data in mm/h in Figure 4 and 5 according to the suggestion made by both referees. However, we present values in $m^3/s$ in Table 2 and Figure 3a as we compare these values to high-flow and low-flow averages in section 3.1.

I find the parameter "DOC load (kg)" largely irrelevant and would remove it together with all the related results and discussions. I would actually change it to "Area specific DOC load (kg $m^{-2}$)", which is a lot more meaningful.

We decided to use the unit [kg/hr], which is a unit typically used for load, instead of [kg]. This will help us to put the number in the context of the specific event and to prevent a bias linked to the duration of the events. We adjusted the related results and discussions. The reference to area is already included in the precipitation specific DOC load [kg $km^{-2}$ $mm^{-1}$]. Moreover, area normalized DOC load values area already mentioned in section 4.3. (L498-499).

While the use of sensors has allowed obtaining high-frequency data, the measurements obtained with sensor loggers are not "continuous" but respond to a fixed-interval. Please, correct the few instances where "continuous measurements" were mentioned and simply specify their frequency or that they were highly-frequent.

We changed the wording according to the reviewer's suggestion.

I would define catchment Markungsgraben as "MG" and catchment Hinterer Schachtencbach as "HS" sooner in the text, and then present them, when possible, always in the same order.

We added the explanation of the abbreviations to section 2.1 (L123) and adjusted the order throughout the manuscript.

Throughout the manuscript, both the term "watershed" and the term "catchment" are used. I would use only one of the two, preferably "catchment".

We changed the wording as suggested.

**Specific comments**

Title

The word "Connectivity" is too vague in the context. I would rather say "hydrological connectivity". I am also a bit sceptical about the word "missing". Maybe a better word is simply "low"? Finally, I would emphasize that the mobilization was studied during rainfall events. What about then: "Low hydrological connectivity during summer controls DOC mobilization and export during rainfall events in a small, forested catchment"? Or something similar.

We changed the title as follows: „Low hydrological connectivity following summer drought inhibits DOC export in a forested headwater catchment"

Abstract

LINE 10. DOC needs to be defined.

We added the definition to the Abstract.

LINE 11. "hypothesized" instead of "hypothesize".

We changed this.

LINE 11. In which contexts is topography a key driver of DOC export? Please, specify (e.g. in headwater catchments).

We added "in headwater catchments" as suggested.

LINE 12. I would rather use "hydrological" instead of "hydrologic", or at least only one of the two terms throughout the paper. Now they appear to be mixed.

We now consistently use "hydrological" throughout the manuscript.

LINE 12. Maybe you better mean "To test this hypothesis"?

We changed this.

LINE 14-16. I don't think this is the best way to describe where the measurements were done. Discharge and DOC were measured in two stream locations, not in a steep hillslope or a flat riparian zone as the sentence as written now implies. Please, rephrase this part to make clear that the measurements were done in the stream, maybe specifying that at one of the locations the stream drains a steep area, whereas at the other location it drains a bigger area that includes a flat and wide riparian zone at lower elevations.

We agree that the description might be misleading and therefore rewrote this part to clarify the location of the measurements (L20-22).

LINE 17. By "During events" you mean during the four studied events? I think so and if so, please specify it.

We added "during the events", which refers to the events mentioned in the sentence before.

LINE 21. This number (522 kg) is largely uninformative without a reference, which in this case I think it should be a normalization to catchment area (see my general comment related to this issue).

As explained above, we prefer to use the reference to a time (the length of the event).

LINE 23. Rather than "lack of hydrological connectivity" I would say "low hydrological connectivity", as the stream is still receiving water from the surrounding catchment area. As I understood, there is no evidence suggesting that the stream is completely disconnected from the catchment under dry conditions, losing water towards the riparian zone (right?). But if there is a complete hydrological disconnection, it should be explained.

We do not have evidence that there is a complete disconnection and therefore toned down the statement by using "low hydrological connectivity" as it will then be used in the title.

LINE 27. I wonder whether there is a better word than "parts" in this context. Maybe "locations" or "compartments"?

We now use the term "sub-catchment" as we already use it in other parts of the manuscript.

LINE 28. Similar to the comment on LINE 23, hydrological connectivity will still occur in the future (unless the stream completely disconnects from the catchment, which I assume it is not the case, not even in summer), only that its degree will be lower depending on the conditions. Thus, I would say something like "will be reduced" or something similar.

We changed this statement to "when hydrological connectivity will be reduced more often".

1 Introduction

LINE 36. Please, move the citation to Drake et al. (2018) to the end of this sentence.

As the citation is related to both sentences, we decided to merge the sentences to one.

LINE 42. The conclusions drawn by Freeman et al. (2001) were admittedly questionable and I would suggest not to cite this paper.

We removed the citation.

LINE 43. "influences terrestrial carbon pools". How? By depleting them? Please, specify.

We changed the sentence to "Rising DOC concentrations indicate an increased leaching from soils and peatlands and have the potential to deplete the terrestrial carbon pools, which are of global importance for carbon storage".

LINE 50. Please, note that a reduction in ionic strength is not an independent process but rather a consequence of a decline in atmospheric acid deposition. Thus, it does not fit in this list.

We moved the reference of Hruška et al (2009) to the other references referring to a decline in atmospheric deposition.

LINE 48-54. In this context, I would suggest having a look at Clark et al. (2010), who nicely summarized the potential factors behind rising DOC concentrations (which have not really changed since that paper was published) and who importantly highlighted that these factors operate on varying temporal and spatial scales. This might be more relevant to your study, although this topic is in general tangential to what it is investigated.

We added a sentence about the difficulty explained by Clark et al. (2010) (L60).

LINE 62. I would write "which can then be mobilized as DOC", rather than " including DOC, which is easily mobilized".

We changed this as suggested.

LINE 72-74. This part of the sentence seems incoherent with respect to the first part of it. Please, rephrase.

We changed the sentence as follows (L80-84): „Anti-clockwise hysteretic loops usually indicate a delayed arrival of DOC at the stream, which can be caused by the source areas being located further away from the stream (Hood et al., 2006; Vaughan et al., 2017), the sources being connected via flow paths with slow transport velocities (Musolff et al., 2017) or by changes in the dominant flow paths and associated changes in hydrological connectivity (Brown et al., 1999; Hagedorn et al., 2000; Schwarze and Beudert, 2009; Strohmeier et al., 2013; Cerro et al., 2014, Ågren et al., 2008; Pacific et al., 2010)."

LINE 79. Please, remove "itself".

We removed "itself".

LINE 79. Does "appears" refer to the beginning of the sentence, i.e. to "Hydrological connectivity". If so, please add commas in between "and therefore […] McDonnel , 2010)".

We added commas after connectivity and response to mark "and therefore runoff and solute response" as an additional information.

LINE 82. I would write "DOC" instead of "C".

We changed this

LINE 89-91. This sentence should be written in past tense, as the hypotheses should define your expectations prior conducting the experiments.

We changed the tenses used here.

LINE 93. I am still not satisfied with the wording "parts of the catchment". Maybe write "between sub-catchments dominated by either of these two topographical configurations", or something similar.

As explained above, we now use the term "sub-catchment".

2 Material and Methods

LINE 105. Maybe it is better to mention here that the Kaltenbrunner Seige sub-catchment was not explicitly studied in this paper. It is also probably better not to mention this catchment again to avoid adding unnecessary unfamiliar names for the reader to keep track.

We now state that the sub-catchment Kaltenbrunner Seige was not studied already in section 2.1. (L125) and reduced the mention of this sub-catchment to the minimum.

LINE 123-130. This information can be presented in a more clear and simplified manner. I would just mention that you have one sampling location close to the outlet of the Markungsgraben catchment at an elevation of 888 m a.s.l., and that this location would be referred thereafter as MG. Briefly say that this catchment is steep and refer to Table 1. Then mention that the second sampling location is close to the outlet of the Hinterer Schachtencbach catchment at an elevation of 771 m a.s.l., and that this location would be referred thereafter as HS. Briefly say that this catchment drains flatter areas with wide riparian zones at lower elevations. I would avoid presenting any other information.

We agree that some of the information in the text was unnecessary and therefore followed the suggestion made by the referee and restructured the paragraph.

LINE 132. At what resolution? Please, specify.

We added the information that all data used for the long-term mean values were measured at a daily resolution (L164).

LINE 134-136. What was this done for? What is the aim of this in the context of the study?

We changed the sentence to: „In order to assess the general meteorological conditions during the sampling period 2018, 2019 and 2020, long-term mean monthly values for the two stations for the period from 1990 to 2010 were calculated." (L162-164). We further explain the characteristics of the sampling periods in section 3.1.

LINE 138-141. The three locations where groundwater level data was monitored should be included in the map of Figure 1. As it is described now, it is difficult to know where they were located with respect to the stream measurement locations. For example, what does "uphill" mean? How far from streams where these three groundwater monitoring stations located, and in which type of soil? In any case, the integration of these data into the story of the paper should be improved. As they are presented now, they do not appear very relevant.

We included the locations in the map of Figure 1 and added the information about the depth of the groundwater wells, which we now name GW1, GW2 and GW3 in order to minimize the usage of complicated original names. We use the groundwater level data to characterize the long-term hydrological conditions of the catchment throughout the sampling periods. As we describe in section 3.1., we can distinguish between two events following dry periods with declining groundwater tables and two events following higher groundwater tables after snowmelt.

LINE 149. What was the resolution of the discharge measurements from the MG site? Given that comparing discharge and exports between the two locations was a major aspect of the study, consideration should be given to the uncertainties associated with the discharge measurements, especially when you have two sources of data with different resolutions. How confident are you that the two discharge time series from the two stream locations can be directly compared?

For both locations, the resolution of the discharge measurements was 15 minutes. We are therefore sure that the discharge time series can be directly compared. We agree that the description was confusing and deleted this sentence. We added the information about the resolution for MG (L178).

LINE 155. So, the grab sample values were added to the software in order to update the internal calibration into a so-called "local calibration", right? This is critical, as I wouldn't trust the default calibration.

We did not add the values to the software by using a "local calibration" but adjusted the default calibration afterwards by using the values measured in the laboratory as we are aware of the fact that the default calibration is not completely reliable. We therefore added: "In order to refine the internal calibration, the DOC concentrations measured…".

LINE 160. Any reason why the DOC calibration for MG was not as good as the calibration for HS?

We are not able to explain why the DOC calibration for MG was not as good as for HS. Due to a technical failure, we had to replace the spectrolyser at MG in July. Therefore, the calibration for the event in September 2020 is different with a $R^2$ of 0.97. This suggests that the mediocre calibration is linked to the specific device. We added the information about the different calibrations at MG to section 2.2.4

LINE 163-165. It feels like this sentence would fit better in the next section. In any case, this part has to be better presented and justified, as it is the basis of all subsequent analyses. Why these four events? What criteria were followed to select them? How do they compare with other events during the study period? Why no other events were included?

As the sampling period was very dry, not many events were available for analysis. Only some small events and very few large events could be observed. Small events led to small discharge and/or DOC responses or no responses at all, which would make the analysis of hysteresis patterns difficult, for instance. We decided to focus on the largest events in order to be able to analyze DOC responses in the stream in detail. We added a sentence about the reasons for selecting the four events presented in the manuscript in order clarify this (L196-198).

LINE 167. For this first sentence to be compelling, first you would need to describe how baseflow was classified. Thus, I would move the sentence to a later point, after you have described how you define events.

We slightly restructured the beginning of this paragraph.

LINE 176. The 15-min resolution values, right? Please, specify it.

Yes, we added the information about the resolution.

3 Results

LINE 196. Please, write "1990-2010" instead of "1990 and 2010".

We changed this.

LINE 197. Do you mean "compared to the long-term average of 1600 mm"?

Yes, we changed this accordingly.

LINE 198-200. I would start the paragraph with this sentence instead.

We changed this as suggested.

LINE 195-200. I wonder how relevant this information and Figure 2 are for the paper. If it is just to put you study period into a long-term context (weather-wise), I would consider removing it, at least the figure. Otherwise, please integrate this part better into the story.

We think that the information is relevant to put the study period into a long-term context, as the study period was particularly dry. However, we agree that the figure is not necessary as the important information is given in the text. We therefore removed the figure.

LINE 204-206. This part related to the groundwater tables (including Figure 3) should also be better integrated into the story. In any case, I am a bit puzzled by what I see in Figure 3. To me it appears that, in general, groundwater tables do not really react to any of the studied events. Is there any reason for this? Where are the groundwater monitoring station located? It seems like soils are very deep there.

The data shows water level in the deep groundwater wells. As it can be seen in Figure 2 (Figure 3 of the first version), the groundwater level varies between 2 and 16 meters below ground. In our opinion, it is therefore not surprising that we observe seasonal variations only instead of a response to events. As explained above, we added the locations of the groundwater wells to Figure 1 and added the information about the depth and geology to section 2.2.2 (L167-171).

LINE 218-221. This part feels like it belongs to the discussion.

We think that this information is useful at this part of the manuscript to explain the reasons for studying the selected events and to highlight the differences between them.

LINE 239-240. I don't know what it is meant here. If you want to refer to the baseflow periods immediately prior the four events, please describe it explicitly.

As this information is not really relevant for the study, we decided to remove this sentence.

LINE 242. "without a clear relation". Did you plot this?

We plotted the relation but decided not to include it in the manuscript in order to focus on other points. However, we added a reference to Table 2, where all values are presented.

LINE 262. "where concentrations decreased soon after reaching the DOC peak". I assume this refers to MG, and not to HS nor to what it is written in parenthesis, but the way the sentence is written makes it confusing. Please, rephrase.

We changed the sentence to: "This resulted in wider hysteretic loops at HS than at MG (larger absolute values of $h$) as the concentrations at MG decreased soon after reaching the DOC peak. "

LINE 283. It is unlikely, but a good theoretical approximation. I would leave this for the discussion, and here just say that you assume equal area contribution.

We prefer to leave the sentence as it is. However, as suggested by Referee #1, we added some more detailed information about how we derived this value to section 2.3. (L218-220).

LINE 290. I realize that the different panels of Figure 4 are not presented in the natural order (a to f) within the results. Could you please either reorganize/relabel the figures or the text to present them in order?

We agree and therefore reorganized the panels.

4 Discussion

LINE 300. But is this driven by P or by $AP_{14}$?

As explained in the following sentence, precipitation alone is not the main driver of Q generation but the antecedent hydrological conditions are of importance.

LINE 306. Please, rewrite this sentence as it is unclear.

We changed the sentence to "To some extent, this observation can be attributed to the larger catchment area contributing to water fluxes at HS, resulting in longer flow pathways and a delayed Q response." and added two references to back up this statement (L357-358).

LINE 312. In which way is hydrological connectivity the driver here? Please, make it explicit at this point, or mention that you will explain it in the following paragraph.

We changed this paragraph as follows: "We suggest that hydrological connectivity between the wide riparian zone and the stream is the major driver for delivering water to the stream. The hydrological connectivity is dependent on both topography and the antecedent wetness conditions as we will explain in the following."

LINE 315. The figure number seems to be missing.

We added the missing figure number.

LINE 321. Please, change "starts sooner" by "is faster".

We changed this as suggested.

LINE 323-324. This needs to be better explained. What kind of "lowlands" and "headwaters" did Zimmer and McGlynn studied and where? Briefly specify it and make the connection to your study.

We rewrote the section about the results found by Zimmer and McGlynn according to the suggestions made by Referee #1 (L371-375).

LINE 332-333. This might be the "expected" range for forested catchments in temperate regions, but it is not the normal range for e.g. boreal, Mediterranean, or tropical sites, so please specify your ecoregion. Also, I would change "expected" by other wording such as "comparable with" or "similar to".

We changed the sentence as follows: "Concentrations were similar to values found in other temperate forested catchments in low mountain ranges."

LINE 333-334. "Larger events generally lead to higher DOC concentrations in streams". Are you referring to your study or to other studies? If the latter, please add a reference. If the former, please remind the reader how you showed this.

We softened the statement and added several references to back it up (L398-399).

LINE 336-339. This is an important conclusion, but it is not universal. To make it more broadly relevant, please argue in what contexts might be applicable.

We agree that we can discuss the broader context in more detail. However, we prefer to do this in the Conclusions section, where we discuss possible implications of climate change for the relative contribution of different sub-catchments. There, we added a sentence regarding the relevance of transport limitation in the context of climate change (L561).

LINE 354-356. Maybe remind the reader that you can make this claim because in this catchment DOC appears to be transport-limited rather than source-limited.

We added the suggestion as follows: "As DOC appears to be transport-limited rather than source-limited, the persistently high concentrations, in combination with a high discharge generation due to the existing hydrological connectivity, could then cause the pronounced DOC export during events following wet antecedent conditions." (L419)

LINE 357-360. I don't know if I agree with the way the transmissivity feedback mechanism is invoked here. The mechanism explains the fast, but deaccelerated increase in groundwater tables due to the saturation of highly conductive shallow soil layers. Thus, at the beginning of an event the increase of groundwater tables would be fast, and then would slow down due to the activation of the highly conductive layers that have a higher lateral water transfer rate. How does the mechanism really connect to your findings? How deep are your soils and how does the groundwater table behave during events? This is where the groundwater table data can be useful.

The groundwater data shown here refer to the deep groundwater level representing slow changes of the groundwater table. They are therefore not helpful to investigate the transmissivity feedback. However, we also installed piezometers in the shallow groundwater of the riparian zone of HS later during the sampling campaign. There we do see the relationship between the groundwater table and discharge as explained by the Referee. We do not include this data as it cannot be linked to the selected events of this study due to different sampling periods and will be part of a different manuscript. Nevertheless, we think that the transmissivity feedback can be of importance in the riparian zone of the lower part of the catchment because there we do see shallow groundwater tables, which can quickly rise into the upper soil layers. This was confirmed by pictures taken by a time-lapse camera in the riparian zone. We added this information to the paragraph about the transmissivity feedback in section 4.1. (L365).

LINE 363. Why later during the event?

During dry periods, those pools are empty and start filling only with the beginning of the precipitation event. They connect to the stream once a certain water level is reached and therefore contribute to discharge later during the event. We therefore changed the sentence as follows: "The possibility that these pools contribute to DOC export when filled with water later during the event is…". (L429)

LINE 366-369. These explanations are crditical in the study, but I am not sure I fully understand them in light of the results. Wouldn't this process imply clockwise hysteresis loops instead of anti-clockwise loops. Why is the activation of sources so slow in your catchment? As I understand it, you are implying that there is a relationship between antecedent wetness and type of hysteresis, but from the data presented in Table 2 and Figure 4b, it doesn't look like there is

a relationship between wetness and "h index" in the HS catchment. This point needs to be carefully addressed.

Anti-clockwise hysteresis loops are caused when the DOC sources in the shallow soil layers first have to be hydrologically connected to the stream. This happens by shallow groundwater tables rising in the soil profile during the rising limb of the event hydrograph. During the falling limb, the DOC rich upper soil layers are still draining while the discharge recedes. The activation of sources is generally slow in our catchment but seems to be accelerating if a certain catchment wetness is reached. It is not unusual to only find anti-clockwise hystereses when comparing our results to other studies. We removed a sentence of section 4.2. in order to clarify this and rewrote parts of section 4.3. as described below (comment concerning Line 410-411).

LINE 370-374. The contrast with other studies in this sense might be also explained by the fact that DOC is transport-limited rather than source-limited, as you argue.

We do not think that the transport-limitation in our catchment is in contrast to other studies as many catchments are transport limited. To further stress the importance of transport limitation, we changed the sentence as follows: "If the soils are wet prior to an event, connected flow paths can quickly be established and DOC transport to the stream occurs faster than during dry conditions, which highlights that DOC export is transport limited in this catchment." (L436).

LINE 410-411. Precisely, as I commented in LINE 366-369, I don't see this pattern in Figure 4b. If I understood it correctly, there might be a weak relationship between catchment wetness and h index for the MG site, but not for the HS site. Is there any type of error in the figure? I might be misunderstanding something, but if the figure and values shown in Table 2 are correct, this part needs to be corrected and some of the discussions you present need to be reconciled with this observation, which is the opposite of what you arguing now.

There was indeed a mix-up of HS and MG in the mentioned sentence, which was misleading. We rectified this and changed the order of sentence to clarify the meaning (L479-483). Although we compare only four data points, we do think that a relation is visible in Figure 4b for MG. We observe smaller hysteresis loops (h closer to zero) during wet conditions than during dry conditions. The event in October, following the dry summer, shows the broadest loop, the event in June the smallest.

LINE 417-419. Where and in what type of catchment did Correa et al. (2019) made this observation.

We added this information: "Correa et al. (2019) made a similar observation in a tropical alpine headwater catchment with anti-clockwise hysteresis..." (L487)

LINE 425. I would end the sentence with "[…] a higher general wetness that favours the build up of DOC in the soil" and would add a reference.

We changed the sentence to: „According to their studies, a flat area would therefore tend to export more DOC than a steep area due to a higher general wetness that favors the build-up of DOC in the soil and hydrological connectivity." (L495)

LINE 429-431. I think this sentence is largely irrelevant and I would remove it.

We removed the sentence as suggested.

LINE 442. "is proportionally higher". Already taking into account the differences in precipitation between the two locations? Please, specify.

We changed the sentence as follows: "The proportional amount of Q arriving from the upper catchment is higher than during the other events; however, the DOC proportion is even higher." (L512-513)

LINE 439-454. The addition of a column in Table 1 with the same information for the entire HS catchment would help interpreting and supporting these explanations.

We agree that the information would be useful and added the column to Table 1 but removed the information about the entire Große Ohe catchment as suggested further down.

LINE 463-464. Which in a way is a specific case of the previous explanation, rather than another different reason. Please, reformulate.

We rephrased the sentence as follows: "A consequence of the reduced connectivity of the lower catchment could be that the riparian pools mentioned above are not connected and thus an important DOC source is not active." (L535-537)

5 Conclusions

LINE 477-479. Do you have reasons to suspect that at this stretch of the stream is a net looser of water at any time of the year? Another reason why DOC can be lost is in-stream mineralization. Can this play a role?

We also investigated possible groundwater gains or losses along the stream using tracer experiments and radon data. The data indicate that exchange with groundwater is especially important in the lower part of the catchment. However, we decided not to include this data as it is beyond the scope of this study. Nevertheless, we will discuss the possibilities of groundwater recharge and contributions in more detail in section 4.1. as suggested by Referee 1 (L471-475). We do not think that in-stream mineralization is of importance as the processes investigated happen at the event-scale and are therefore rather fast in contrast to mineralization processes. We added this information to the conclusion (L550).

Tables and Figures

Table 1. I would remove the column with the information about the entire Grosse Ohe catchment, as it is more distracting than anything else. As I understand, the information presented for the Hinterer Schachtencbach catchment only reflects the local sub-catchment, but I would also like to see the analogous, integrated information for the entire catchment (i.e. for the whole 3.5 km$^2$ that include the other two subcatchments).

As explained above, we added the column to Table 1 but removed the information about the entire Große Ohe catchment as suggested.

Figure 5. Please, add HS and MG on top of the left and right panels, as in Figure 6. Please change to "mm/15 min" the units of the precipitation (if that's the case).

We added HS and MG as suggested and added the units of precipitation (mm/h).

Figure 6. Besides the colour code, a small arrow (or a couple of arrows in the lower panels) indicating the direction of the hysteresis loops in each panel would help visualizing and interpreting the results.

We added arrows as suggested.

Suggested references

Clark, J. M., Bottrell, S. H., Evans, C. D., Monteith, D. T., Bartlett, R., Rose, R., Newton, R. J., and Chapman, P. J.: The importance of the relationship between scale and process in understanding long-term DOC dynamics, Science of the Total Environment, 408, 2768-2775, 10.1016/j.scitotenv.2010.02.046, 2010.

Katharina Blaurock

On behalf of all co-authors

---

## Author Response (AR2)

Dear editor,

we are pleased to read that you and the referees think that the manuscript has improved as a result of the revisions. We appreciate the additional comments of both referees during this second review. In the following, we will address all comments and explain which changes were made. In addition to the suggestions made by the referees, we made some minor changes, which we explain following the referees' comments. We refer to the line numbers of the marked up manuscript.

**Referee #1 (Report #2)**

Summary
I reviewed a previous version of this manuscript and have read through the response to reviewer document, track changes document, and the newly revised manuscript. The authors did a thorough job responding to the two referees, who had some overlapping comments and suggestions. These revisions greatly improved the quality and clarity of the manuscript. The authors addressed concerns about sample size, explanatory variables, and other clarifying requests. While I don't necessarily agree with all of the authors' final interpretations (e.g. season is not a critically important control on DOC export), I find that the authors rationalize their findings and interpretations in a convincing way and support their claims well with an up-to-date literature review. Overall, I believe the manuscript is suitable for acceptance at HESS pending some minor revisions, as described below.

Minor
I find the manuscript to be convoluted since the authors are trying to discuss a range of explanatory variables for two catchments over four events. I find a manuscript to be most impactful when it is easily accessible to readers, but as the paper is currently presented there is a lot left for the reader to untangle. Thus, one strong suggestion I have for the authors is to include a conceptual figure at the end of the manuscript that visually summarizes the main key points in terms of controls on DOC export.

We created a conceptual figure (Fig. 7), which summarizes the link between antecedent wetness conditions, event size, topography and DOC export. We refer to this figure in section 4.2. (L. 426) and section 4.3. (L. 493).

I appreciate the changes to the title of the manuscript to try to clarify the message. However, I find the current title still doesn't capture the main objective of the paper. That is because not all of the events occurred during summer, yet this title suggests the study is focused in summer. I would recommend clarifying the title to reflect the range of event timing that was included in this study.

We do understand the referee's wish to present the main objective of the paper in the title. However, our reason for choosing this title is the goal to highlight the main finding of the paper. Therefore, we prefer to leave the title generally as it is. However, we decided to exchange "during" with "after" as the observed events end the summer drought.

The revised abstract reads very well and the minor changes to describe the two catchments helped clarify the site specifics.

L 39 – need comma after "5.1 Pg C"

We added the comma.

L 57 – missing period at the end of the sentence

We added a period.

Figure 1 – It is still unclear how the authors used the 5 m DEM to identify stream channels. Did they use a contributing area threshold, or base it topographic proxies, or something else? Our understanding of the stream network is important for understanding DOC export behavior as it can help us understand the aquatic connection to the terrestrial landscape, thus I think it is important to be clear.

The stream channels used in Figure 1 are based on the official shape files provided by the Bavarian State Office for Environment, which had identified the stream channels using topographic proxies and calculating contributing areas. We added this information to the caption of Figure 1.

L 218 – change Figure 23 to correct Figure reference (Figure 2).

We changed the Figure references.

L 253 – I would not classify event-scale measurements as "long-term"

In this sentence, we do not refer to the event-scale measurements but intend to present the DOC concentrations found during baseflow, which we measured during our long-term measurements. Therefore, we decided to leave the sentence as it is.

L 313- The events were also characterized by contrasting seasonality.

In section 4.1., we do not yet discuss processes involving DOC. Therefore, we mention the antecedent wetness conditions only rather than discussing seasonality at this point as well as this would include other factors (e.g. temperature), which indeed do influence DOC production but not the processes involving discharge only. We do discuss seasonality in section 4.2., where we focus on DOC mobilization processes.

L 412 – Wouldn't an event in May have a different seasonal effect than an event in September in terms of biological activity, not just wetness conditions? May is following a wetter dormant season and September is following a dry growing season. One would expect this seasonal difference to influence DOC availability. While the authors mention 'warm summer months' in the paragraph, it is buried in the paragraph and should be included in the topic sentence at least.

We added the information about the contrasting seasonality more prominently.

L 480 – add period after "conditions"

We added a period.

Conclusion - I am still not fully convinced that season is not an important contributor to DOC export differences seen across events – perhaps the authors could add one sentence in the conclusions that state why event size, antecedent wetness, and topography are important, but season is not.

We included a sentence about the effect of seasonality in the conclusions (L520).

**Referee #2 (Report #1)**

Blaurock et al. sent a revised version of their previously submitted manuscript, together with their replies to reviewer comments. In this study, they investigated the mobilization of DOC during storm events in two topographically contrasting forest catchments and concluded that event size, antecedent wetness conditions, and topography are major determinants of DOC mobilization and export.

I carefully read the response letter and the revised manuscript, which I enjoyed. I thank the authors for addressing all my comments and making the pertinent changes in the manuscript or explaining why they didn't in those cases where they disagreed with my points. For the most part, I am satisfied with the responses and the changes made in the manuscript, which I believe have helped increasing its clarity. I do however have a few further comments that follow up on a few of the discussed topics and that I would like to see considered/clarified before a final version of the manuscript can be accepted.

I understand now that the groundwater tables plotted in Figure 2 are from three deep wells and that they are used as a proxy for some kind of overall wetness conditions in the catchment that follow seasonal patterns rather than event-dynamics. However, the data from these deep wells should not be confused with groundwater table variations occurring in the shallow soil, which do show small-temporal scale dynamics responding to events and which you also discuss at times (e.g. L. 323-327 and L. 335-339). The distinction between these two types of groundwater tables has to be more clearly made, for example by explicitly differentiating between deep groundwater tables (with seasonal dynamics) versus shallow groundwater tables (which responds to precipitation inputs and relate to water and solute delivery during events). The way groundwater tables are presented now leads to confusion.

We now write "deep groundwater" in section 3.1., where we use this data as a proxy for the wetness conditions. Moreover, we specify this in the title of section 2.2.2., where we explain, which data was used as "deep groundwater". In section 4.1., we talk about groundwater in a more general way or specify "shallow groundwater" when necessary.

Additionally, I wonder if a further distinction can be made by describing the water storage structure in the catchment. The way you describe things makes me think that the catchment has a perched shallow aquifer that overlies a saturated deep aquifer with varying degrees of connectivity to the stream. Is this the case? If so, please specify it and link it to the deep versus shallow groundwater table distinction.

As we have only limited information on shallow groundwater levels, we cannot discuss processes involving deep and shallow groundwater in detail. In our opinion, we do lack the data to conclude that the catchment shows a perched shallow groundwater later and overlies a deep aquifer as you suggest, although this definitely could be a possibility. However, the deep groundwater wells are also located at different locations than the other sampling site and it is therefore difficult to compare the data.

While I tend to believe that in-stream DOC mineralization might not be significant during rainfall events that produce high water velocities and lead to short water residence times, this paradigm might not be as universal as previously thought as there are recent studies suggesting the opposite might be true in certain settings (e.g. Bernal, S., Lupon, A., Wollheim, W. M., Sabater, F., Poblador, S., & Martí, E. (2019). Supply, Demand, and In-Stream Retention of Dissolved Organic Carbon and Nitrate During Storms in Mediterranean Forested Headwater Streams. Frontiers in Environmental Science, 7, 14. doi:10.3389/fenvs.2019.00060). Therefore, consider this information and back up your statement in L. 499-500 with some references.

We added additional information about in-stream mineralization to the conclusion including references.

L. 91. Do you mean "hydrological conditions" here, or rather "(antecedent) wetness conditions"?

We changed this to "antecedent wetness conditions".

L. 101-102. For coherence, I think it is better to write "antecedent wetness conditions" here.

We changed this to "antecedent wetness conditions".

L. 163. Please, remove "continuously".

We removed "continuously".

L. 178-179 and L. 218. Please, correct figure numbers.

We corrected the figure numbers.

L. 235. What is "HQ1"? Please, consider removing if not relevant.

We removed the HQ1 information and only mention the mean high-flow discharge.

L. 500-503. Check this sentence, it includes the same information twice.

We adjusted the sentence in order to avoid duplicates.

**Additional minor changes**

Fig. 3. We adjusted the limits of the y-axis of Fig. 3e and therefore inserted a new version of Fig. 3.

L. 454. We moved the sentence about the comparison of the events in June 2018 and September 2020 to L. 460.

Katharina Blaurock

On behalf of all co-authors